# SceneCOT: Eliciting Grounded Chain-of-Thought Reasoning in 3D Scenes

**Xiongkun Linghu**[1], **Jiangyong Huang**[1,2], **Ziyu Zhu**[1,3], **Baoxiong Jia**[1,†], **Siyuan Huang**[1,†]

[1]State Key Laboratory of General Artificial Intelligence, BIGAI
[2]Peking University, [3]Tsinghua University

## Abstract

Existing research on 3D Large Language Models (LLMs) still struggles to achieve grounded question-answering, primarily due to the under-exploration of *the mechanism of human-like scene-object grounded reasoning*. This paper bridges the gap by presenting a novel framework. We first introduce a grounded Chain-of-Thought reasoning method in 3D scenes (SceneCOT), decoupling a complex reasoning task into simpler and manageable problems, and building corresponding visual clues based on multimodal expert modules. To enable such a method, we develop SceneCOT-185K, the first large-scale grounded CoT reasoning dataset, consisting of 185K high-quality instances. Extensive experiments across various complex 3D scene reasoning benchmarks demonstrate that our new framework achieves strong performance with high grounding-QA coherence. To the best of our knowledge, this is the first successful application of CoT reasoning to 3D scene understanding, enabling step-by-step human-like reasoning and showing potential for extension to broader 3D scene understanding scenarios.

## 1 Introduction

Understanding 3D scenes is a fundamental capability for building human-level embodied agents (Chen et al., 2019; Yang et al., 2025b; Chen et al., 2024a; Gong et al., 2023; Song et al., 2025). Despite growing interest and progress in this area (Chen et al., 2024c; Jia et al., 2024; Zhu et al., 2024b; Huang et al., 2024b; Chen et al., 2023b; Zhou et al., 2024), reasoning in complex 3D environments remains highly challenging, especially in embodied and situated settings (Ma et al., 2023; Linghu et al., 2024). Reasoning in 3D scenes requires navigating large spaces, interpreting intricate spatial relations, and coping with partial observability. Addressing these challenges calls for principled task decomposition and step-by-step reasoning, yet existing research has largely overlooked this aspect.

Recent benchmarks highlight the consequences of this gap. In particular, Beacon3D (Huang et al., 2025a) reveals that current 3D vision-language models often produce plausible answers without grounding them in the scene, leading to poor grounding-QA coherence. This underlines a fundamental limitation: while models may generate fluent responses, they fail to connect intermediate grounding steps with the final reasoning outcome. We argue that achieving human-like 3D reasoning requires answers that emerge from transparent, grounded, and stepwise reasoning processes.

In the language domain, Chain-of-Thought (CoT) reasoning (Wei et al., 2022) has demonstrated the power of step-by-step problem decomposition, enabling large language models to achieve superhuman performance in math, science, and logic tasks (Wei et al., 2022; Guo et al., 2025; Jaech et al., 2024). CoT mirrors human cognition by breaking complex problems into manageable subproblems, a paradigm that naturally aligns with the multi-hop reasoning needed in 3D scenes. Yet, directly transferring CoT to 3D settings is nontrivial due to the difficulty of aligning language-based reasoning with multimodal 3D scene representations. To date, CoT has been explored in 2D vision-language reasoning (Shao et al., 2024a; Liu et al., 2025), leaving its potential in 3D reasoning largely untapped.

---

[†]Corresponding author.

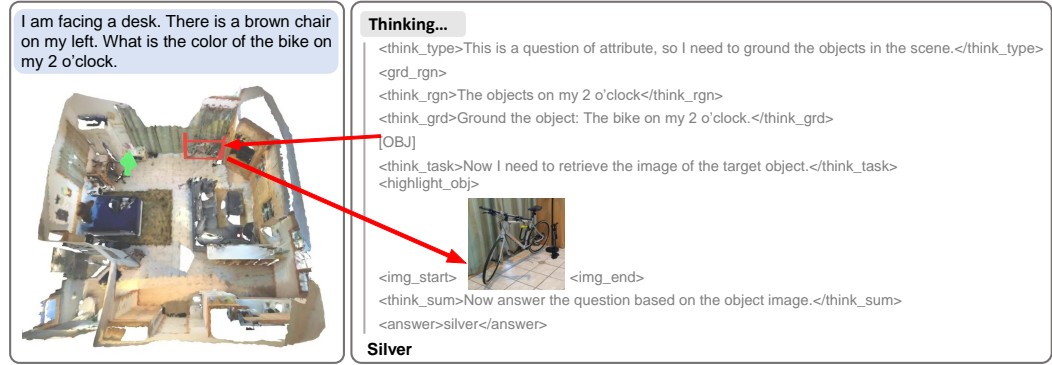

Figure 1: **Reasoning chain visualization of SCENECOT**. Example of how SCENECOT decomposes a 3D question into step-by-step reasoning: from identifying the task type, localizing relevant objects, and grounding the target entity, to retrieving the visual clue and generating the final answer.

In this work, we present SCENECOT, a novel framework for step-by-step grounded reasoning in 3D scenes. SCENECOT explicitly decomposes complex 3D reasoning tasks into four stages: (1) task recognition and analysis, (2) task-relevant region localization, (3) entity and attribute grounding with multimodal expert modules, and (4) grounded reasoning that integrates intermediate results into coherent final answers. This hierarchical workflow ensures that each answer is supported by explicit grounding steps, thereby enhancing grounding-QA coherence. To enable such reasoning, we construct SCENECOT-185K, the first large-scale grounded CoT dataset for 3D reasoning, containing over 185K high-quality reasoning traces across diverse 3D QA benchmarks. Each trace captures the full stepwise reasoning trajectory, including task-oriented region selection, object grounding, and final answer generation. Extensive experiments on situated reasoning benchmark MSQA (Linghu et al., 2024), as well as grounding-QA coherence evaluation on Beacon3D (Huang et al., 2025a), show that SCENECOT achieves strong performance while producing reasoning that is both interpretable and faithfully grounded.

In summary, our contributions are as follows:

- We propose SCENECOT, a novel Chain-of-Thought reasoning framework that decomposes complex 3D scene reasoning tasks into manageable steps, enabling human-like, grounded, and interpretable reasoning.
- We construct SCENECOT-185K, the first large-scale dataset with stepwise grounded reasoning traces in 3D scenes, containing over 185K high-quality instances.
- We demonstrate that SCENECOT achieves strong performance on challenging 3D reasoning benchmarks, and importantly, improves grounding-QA coherence, as revealed by in-depth analyses on Beacon3D.

## 2    RELATED WORK

**LLMs for 3D Scene Understanding**    Understanding 3D scenes is essential for developing human-like intelligence, and recent studies have increasingly explored the use of LLMs for 3D vision-language understanding. Early efforts such as 3D-LLM (Hong et al., 2023) lift multi-view 2D features into 3D space and align them with text embeddings, while PointLLM (Xu et al., 2023) leverages point cloud encoders for object-level geometry reasoning. LEO (Huang et al., 2024b) aligns object-centric 3D representations with LLMs for 3D vision-language (3D-VL) tasks. Subsequent works have expanded this line of research (Chen et al., 2024b; Yang et al., 2025a; Fu et al., 2025; Chu et al., 2024; Zhang et al., 2024c; Qi et al., 2023). For instance, Grounded 3D-LLM (Chen et al., 2024c) enhances point-level semantic alignment with language, Chat-Scene (Huang et al., 2024a) builds per-object 3D features via mask proposals, LLaVA-3D (Zhu et al., 2024a) integrates 3D positional embeddings into LLaVA (Liu et al., 2023), Video-3D LLM (Zheng et al., 2024) leverages pretrained video MLLMs for temporal-spatial context, and SplatTalk (Thai et al., 2025) applies 3D Gaussian Splatting (Kerbl et al., 2023) to generate language-aligned 3D tokens. LEO-VL (Huang et al., 2025b) further improves the efficiency of scene representation for 3D MLLMs. Despite these advances, most 3D-LLMs rely on sparse supervision in an end-to-end training paradigm, with limited exploration of intermediate reasoning processes. As shown in Beacon3D (Huang et al., 2025a), these models

often produce answers that appear correct but lack explicit links to scene grounding, leading to poor grounding-QA coherence. This reveals a fundamental gap: while 3D-LLMs can encode multimodal representations, they rarely incorporate structured, step-by-step reasoning mechanisms that are essential for robust 3D scene understanding. We argue that Chain-of-Thought (CoT) reasoning remains largely untapped in 3D, but is necessary for overcoming overfitting and building more interpretable, human-like 3D-VL models.

**Reasoning Capability of LLMs and MLLMs**   LLMs have demonstrated remarkable reasoning abilities across diverse domains such as mathematics, programming, and scientific QA (Guo et al., 2023; Ou et al., 2024; Chen et al., 2024d; Huang et al., 2022; Shao et al., 2024b; Jaech et al., 2024). These capabilities are significantly enhanced by Chain-of-Thought (CoT) prompting (Wei et al., 2022), which decomposes complex tasks into step-by-step subproblems. Inspired by this, researchers have sought to extend reasoning into multimodal settings. Flamingo (Alayrac et al., 2022) bridges frozen vision and language backbones with cross-attention, Shikra (Chen et al., 2023a) and KOSMOS-2 (Peng et al., 2023) emphasize grounded reasoning over visual inputs, while OMG-LLaVA (Zhang et al., 2024a) unifies pixel-, object-, and region-level understanding. More recent works attempt to bring step-by-step reasoning into MLLMs. V∗ (Wu & Xie, 2024) incorporates sequential visual search for fine-grained recognition, Video-of-Thought (Fei et al., 2024) introduces a perception-to-recognition workflow for video reasoning, and Visual CoT (Shao et al., 2024a) explicitly integrates bounding boxes and patch-level grounding as intermediate "thoughts." Commercial models such as GPT-o3 and GPT-o3 mini (OpenAI, 2025) showcase impressive visual reasoning abilities by combining multiple visual experts. However, these advances remain confined to 2D inputs. They lack explicit grounding in 3D space, limiting their ability to reason about spatially complex, embodied scenarios. In summary, while CoT reasoning has transformed both text-only and 2D multimodal models, it has not yet been fully explored in 3D scene understanding. Bridging this gap requires designing frameworks that integrate step-by-step grounded reasoning with explicit grounding-QA coherence, which is the focus of our work.

## 3   SCENECOT: STEP-BY-STEP REASONING IN 3D SCENES

In this section, we present the design of SCENECOT, beginning with the formalization of step-by-step reasoning traces in common 3D reasoning tasks (Sec. 3.1). This structure reflects the human problem-solving process for complex understanding, as discussed in Sec. 1. We then describe the detailed learning and inference pipeline of SCENECOT, illustrating how it integrates and leverages 3D-CoTs to enhance reasoning capabilities in 3D scenes (Sec. 3.2).

### 3.1   CHAIN-OF-THOUGHTS IN SCENECOT

Given a 3D scene, an agent's situation, and a question to be answered, we define the reasoning trace for answering the question as a concatenation of the following step-by-step descriptions:

1. **Task Recognition and Analysis:** The reasoning trace begins with the identification of the underlying task required to answer the question (*e.g.*, counting, navigation) along with initial analysis for solving it (*e.g.*, ground objects for counting). This information is critical for the subsequent reasoning process. It may also determine which specialized models (*e.g.*, detection, segmentation) to invoke in subsequent steps. We encapsulate this task-level guidance using the special token `<think_type>` within the reasoning trace.
2. **Task-relevant Region Localization:** Based on the task hints in the question and the agent's situation context, we significantly reduce the reasoning space by first providing region-level grounding for localizing the task-relevant subregion of the scene. Specifically, depending on the question, we discretize the surrounding space using either directional clues (*e.g.*, left, right, front, back) or a clock-based reference frame (*e.g.*, 1-12 o'clock). The sentence enclosed by `<think_rgn>` indicates the candidate region for the symbolic engine to parse the corresponding objects.
3. **Entity Grounding:** This step grounds the target objects relevant to answering the question. We first generate detailed grounding instructions that encode object semantics, attributes, and relational context. These instructions are enclosed by `<think_grd>`, and followed by a special `[OBJ]` token, which serves as a trigger for invoking specialized grounding modules to localize the referenced object(s).

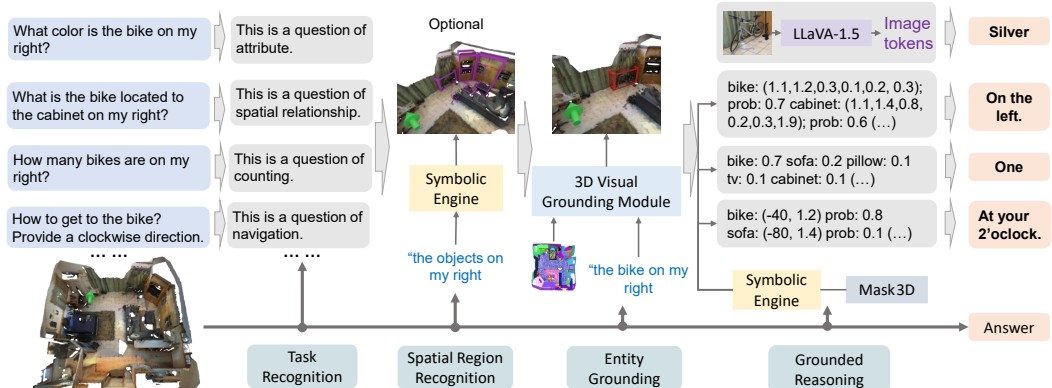

Figure 2: **SCENECOT framework**. The model decomposes 3D scene reasoning into four steps: task recognition, spatial region recognition, entity grounding, and grounded reasoning. Each stage introduces explicit grounding signals (*e.g.*, objects, attributes, spatial positions), ensuring step-by-step reasoning and improved grounding-QA coherence.

4. **Grounded Reasoning:** Given the candidate object(s), we generate task-specific instructions, enclosed by `<think_task>`, that specify what information regarding these objects is necessary for downstream reasoning. These instructions guide the grounding model to retrieve or compute relevant information based on task requirements, such as retrieving 2D images for attribute recognition and exporting object class probabilities for object existence verification. We consider the following types of grounded object information, each annotated with distinct tags:

   - *Object probability* (`<obj_prob>`): For tasks such as counting and existence verification, we record the class probabilities of grounded objects, indicating the presence of objects.

   - *3D Object Location with Probability* (`<obj_loc_prob>`, `<obj_loc_plr_prob>`): For tasks that require spatial reasoning, we record object class probabilities along with their positions in 3D space(`<obj_loc_prob>`). For navigation-related tasks, we represent object locations in a 2D polar coordinate (`<obj_loc_plr_prob>`) frame to facilitate direction-based reasoning.

   - *Object Image Tokens* (`<highlight_obj>`, `<img_start>`, `<img_end>`): For fine-grained attribute description tasks, we use `<highlight_obj>` to trigger image retrieval, inserting object-level image patches as visual tokens tagged by `` to support appearance-based reasoning.

After obtaining the grounded object information necessary for solving the task, we include a summarization hint marked with `<think_sum>` to guide the reasoning process, followed by the generation of the final answer, tagged with `<answer>`. An illustrative walkthrough of a sample reasoning trace for a situated question answering task from the MSQA dataset is shown in Fig. 1, with further details on each sub-process of the reasoning trace discussed in Sec. A.

## 3.2 SCENECOT LEARNING AND INFERENCE

We provide an illustrative explanation of how SCENECOT learns and performs inference with our defined 3D-CoTs in Fig. 2. At its core, SCENECOT is built upon a powerful Multi-modal LLM (MLLM), which serves as the primary reasoning engine. To support the step-wise reasoning structure introduced in Sec. 3.1, we incorporate two types of modular components:

- Specialized 3D-VL and 2D-VL models are employed for entity grounding and image reasoning. These models are initialized with pre-trained weights and jointly updated during the training of SCENECOT. In particular, a 3D visual grounding model and a 2D vision-language model serve as the backbone for the grounded reasoning process.

- Symbolic engines, including off-the-shelf parsers and pre-trained models, are used to extract object-specific grounding information (e.g., location, coordinates, image patches) to support region recognition and grounded reasoning, as described in Sec. 3.1. These models remain fixed and are not updated during SCENECOT training. The contextual inputs are constructed through a predefined programming procedure, with implementation details provided in Sec. A.

To train SCENECOT, we use a dataset of annotated reasoning traces as described in Sec. 3.1, jointly optimizing the reasoning engine and grounding modules under the following objective:

$$\mathcal{L} = \mathcal{L}_{\text{CoT}} + \mathcal{L}_{\text{ans}} + \mathcal{L}_{\text{ground}}, \tag{1}$$

where $\mathcal{L}_{\text{CoT}}$ and $\mathcal{L}_{\text{ans}}$ are causal language modeling losses for predicting the reasoning trace and final answer, respectively. $\mathcal{L}_{\text{ground}}$ is a cross-entropy loss applied only to the specialized grounding module for accurate object grounding. We train the MLLM model using LoRA.

In the inference stage, SCENECOT follows the reasoning steps specified by the predicted 3D-CoTs, invoking the appropriate modules to generate the final answer. Specifically, we use Mask3D (Schult et al., 2022) to an initial set of object proposals for the specialized grounding model to select. For special tokens that invoke function calls, the corresponding modules are executed externally, and their outputs are concatenated with prior predictions and fed back into SCENECOT for a new inference pass to complete the reasoning process. Additional details on model inference are provided in the Sec. A.4.

## 4 THE SCENECOT-185K DATASET

To enable the learning of SCENECOT, we develop a large-scale 3D-CoTs dataset, SCENECOT-185K, containing 185K data instances to support step-by-step reasoning in 3D scenes. The dataset comprises two representative tasks in 3D scene reasoning: (1) Situated Reasoning and (2) Object-Centric Reasoning. It follows the standard 3D-CoTs structure as defined in Sec. 3. We construct the dataset through a two-step process: metadata collection and reasoning trace generation.

### 4.1 METADATA COLLECTION

For Situated Reasoning, we use MSQA as the data source. Following the definition in Sec. 3.1, we collect the following components: (1) object instances within the corresponding sub-region, (2) the question type, and (3) the grounding text. For question types, we adopt the primary categorization defined in MSQA and construct the corresponding COT data using the official metadata. As Beacon3D emphasizes grounded reasoning, we treat it as part of the *Attribute*/*Description* sub-tasks within our unified task space.

**Region-relevant and Question-relevant Objects Extraction**  First, we design a rule-based procedure to extract target objects based on the agent's location and orientation within the 3D scene. This begins by parsing the question and extracting directional cues using regular expression matching. In MSQA, directional information typically falls into two categories: cardinal directions (left, right, front, back) and clock-based directions (*e.g.*, "at the 1-12 o'clock"). For instance, given the question "How many tables are on my right?", we extract all objects located to the right of the agent. This rule-based method ensures that answers can be accurately inferred from the object list within the corresponding sub-region. Secondly, for the target object entities, we design a rule-based method for *Existence*/*Counting* to ensure the correctness. For the remaining sub-tasks, we inherit the official annotations of the target objects in the released data.

**Data Generation of GQA3D**  For Beacon3D, we cannot directly construct the COT data due to the absence of metadata. To obtain a high-quality training set, we leverage Nr3D as our metadata source and construct a new grounded question answering dataset, GQA3D. Nr3D is a 3D Visual Grounding benchmark that primarily provides grounding texts and the IDs of target objects. We use GPT-4o to generate QA pairs based on the corresponding object images. All generated QA pairs are categorized as *Attribute* type in SCENECOT-185K. Implementation details are provided in Sec. B.2.

### 4.2 REASONING TRACE GENERATION

After collecting the metadata, we construct the full reasoning chain following the steps outlined in Sec. 3.

1) For the sub-tasks *Counting*, *Existence*, *Refer*, *Room Type*, and *Affordance*, we generate ground-truth thoughts using semantic labels and pseudo probabilities. Specifically, we randomly assign values between 0.5 and 1.0 to represent target objects, while assigning values between 0 and 0.5 to non-target

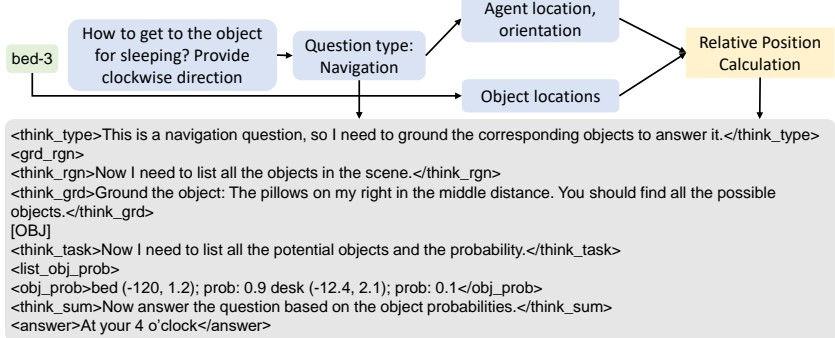

Figure 3: **Illustration of reasoning trace generation**. An example of a navigation question and its corresponding reasoning trace. The process involves identifying the question type, grounding relevant objects, computing relative positions based on agent location and orientation, and generating the final answer through step-by-step reasoning.

objects. To control the token length of the input prompts, we cap the number of objects per training instance.

2) For the *Spatial Relationship* sub-task, we compute relative coordinates using the agent's location, orientation, and the positions of objects under the 3D rectilinear coordinate system.

3) For the *Navigation* sub-task, object locations are represented in a 2D polar coordinate system. To support the *Attribute* and *Description* sub-tasks, which require image retrieval, we construct an object image library. Object images are extracted from RGB frames in ScanNet using object positions and camera poses. We illustrate a representative reasoning trace generation pipeline of *Navigation* in Fig. 3. We finally get 145.6K and 40K data instances for Situated Reasoning and Object-Centric Reasoning, respectively. We also conducted a manual check of data quality. The statistical results can be found in Sec. B.3.

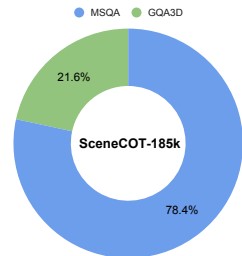

Figure 4: **Data distribution of SCENECOT-185K**.

## 5 EXPERIMENTS

### 5.1 EXPERIMENTAL SETUP

**Tasks and Data.** We evaluate SCENECOT on two representative 3D scene reasoning tasks: Situated Reasoning (Situated Reasoning) and Object-Centric Reasoning (Object-Centric Reasoning). For Situated Reasoning, we adopt the MSQA benchmark built on ScanNet, following the evaluation protocol of (Thai et al., 2025). To ensure fair and consistent comparison, we use the refined Version-2.1 of MSQA (Linghu et al., 2024), and re-implement MSR3D and GPT-4o on this version. For MSR3D, we follow the official setup, which includes the merged training data and object mask proposals from Mask3D. For Object-Centric Reasoning, we evaluate on the Beacon3D benchmark (Huang et al., 2025a), which explicitly measures grounding-QA coherence. Following the official settings, we use ground-truth object masks for all methods to isolate reasoning capability from detection performance.

**Evaluation Protocols and Baselines.** On both MSQA and Beacon3D, we use GPT-score as the primary evaluation metric. For MSQA, models jointly predict object masks and textual answers, while for Beacon3D, evaluation is performed using ground-truth masks. Beacon3D further reports two complementary metrics: (1) GPT-Score (by case), which averages scores across individual QA pairs, and (2) GPT-Score (by object), which requires all QA pairs associated with an object to be correct, directly reflecting grounding-QA coherence. In addition, we analyze the grounding-QA coherence metrics provided in Beacon3D to highlight the improvements brought by our method. We compare SCENECOT against a broad set of 3D scene reasoning baselines. On Beacon3D, we focus on object-centric baselines that explicitly incorporate grounding, including GPT-4o (OpenAI, 2023), MSR3D (Linghu et al., 2024), PQ3D (Zhu et al., 2024b), Chat-Scene (Huang et al., 2024a), and SceneVerse (Jia et al., 2024). On MSQA, we additionally include SplatTalk (Thai et al., 2025), which

Table 1: **Experimental Results on MSQA and Beacon3D**. *: GPT-4o's input contains ground-truth object labels, locations, and attributes. ‡: The result of MSQA is not based on Version-2.1 data. MSR3D, Chat-Scene, and LEO are trained on SCENECOT-185K-QA(no grounded COT). †: The models are trained on our dataset. The **best** and second-best performances are highlighted across the entire table. In the third column, 'Grounded' indicates whether the reasoning results can be explicitly linked to specific entities.

| Methods | Grounded? | MSQA | | | | | | | Beacon3D | |
| | | Count. | Exist. | Attr. | Spatial | Navi. | Others | Overall | Case | Obj. |
|---|---|---|---|---|---|---|---|---|---|---|
| GPT-4o* | ✗ | 32.3 | 79.3 | 79.0 | 37.0 | 31.7 | **91.6** | 52.3 | 57.1 | 20.2 |
| LEO | ✗ | 32.5 | 88.5 | **58.7** | 44.2 | 39.6 | 81.4 | 54.8 | 43.2 | 7.8 |
| MSR3D | ✗ | 32.3 | **93.1** | 50.0 | 46.5 | 54.1 | 75.6 | 54.2 | – | – |
| Chat-Scene | ✗ | – | – | – | – | – | – | – | 45.8 | 7.8 |
| SplatTalk‡ | ✗ | 19.6 | 60.3 | 44.0 | 35.8 | 35.5 | 61.8 | 41.8 | – | – |
| LEO† | ✗ | 29.3 | 87.5 | 55.1 | 46.3 | 45.6 | 81.4 | 52.9 | 52.5 | 12.7 |
| MSR3D† | ✗ | 32.7 | 87.5 | 53.7 | 44.3 | 51.5 | 72.3 | 52.8 | 51.4 | 11.9 |
| Chat-Scene† | ✗ | 37.4 | 92.0 | 49.0 | 47.0 | **58.3** | 83.7 | **56.6** | 53.6 | 14.0 |
| SCENECOT | ✓ | **47.9** | 82.1 | 49.6 | **47.2** | 51.6 | 80.3 | 55.6 | **58.9** | **23.2** |

Table 2: **Grounding-QA Coherence comparison across methods**. Main metrics: GC: good coherence (both grounding and QA correct); QA (Obj.): per-object QA performance. Additional reference metrics: Type 1: grounding correct but QA wrong; Type 2: QA correct but grounding wrong; DF: double failure (both wrong); $R_1$ = Type1 / (Type1 + DF); $R_2$ = Type2 / (Type2 + GC).

| Method | Grounded? | GC ↑ | QA (Obj.) ↑ | DF ↓ | Type 1 ↓ | Type 2 ↓ | $R_1$ ↑ | $R_2$ ↓ |
|---|---|---|---|---|---|---|---|---|
| LEO | ✗ | 1.6 | 7.8 | 2.2 | 55.2 | 40.9 | 3.7 | 96.2 |
| PQ3D | ✗ | 16.5 | 3.5 | 40.6 | 31.9 | 10.8 | 56.6 | 39.7 |
| SceneVerse | ✗ | 20.4 | 6.6 | 31.6 | 28.3 | 19.5 | 50.6 | 48.5 |
| Chat-Scene | ✗ | 19.5 | 7.8 | 24.7 | 29.8 | 25.8 | 44.4 | 56.9 |
| SceneCOT | ✓ | **34.7** | **23.2** | 16.8 | 24.1 | 16.8 | 58.9 | 41.0 |

reports results on this benchmark. To ensure fairness, we retrain several baselines on our training set and report the updated results.

**Implementation Details.** We build SCENECOT based on LLaVA-1.5, an open-source multimodal LLM framework built upon Vicuna-7B. For the *Attribute* and *Description* sub-tasks, the selected object image is passed through a 2D vision encoder. The resulting image feature is then projected into the language embedding space via learnable projection layers. During training, we freeze these projection layers and apply LoRA to fine-tune the parameters of the LLM. For 3D visual grounding, we adopt a fine-tuned version of PQ3D as our expert model. This model has been trained on a subset of the domains provided in SceneVerse (Jia et al., 2024). Additionally, we design a lightweight object mask predictor to estimate object logits. During training, we fine-tune both the parameters of PQ3D and the object mask predictor. Our model is trained for 5 epochs using 4 A100 GPUs. We provide more implementation details in Sec. A.5.

## 5.2 MAIN RESULTS ON 3D QA AND GROUNDING-QA COHERENCE

In the main results, we evaluate both 3D QA performance and grounding-QA coherence, and compare SCENECOT with baseline methods.

**SCENECOT demonstrates strong QA performance on Situated Reasoning and Object-Centric Reasoning**. It achieves notable gains on the challenging *Counting* sub-task by explicitly enumerating objects in the relevant sub-region based on semantic similarity, providing clear visual grounding for reasoning. SCENECOT also performs well on *Spatial* tasks, including *Refer* and *Spatial Relationship*. Its performance is lower on *Existence* and *Attribute* compared to LEO and MSR3D, mainly due to grounding errors (see Sec. 5.3). While slightly below Chat-Scene overall, SCENECOT substantially outperforms all baselines on Beacon3D thanks to its scene-object grounded reasoning. In contrast, Chat-Scene directly predicts answers from question and object-centric tokens without explicit grounding, limiting its ability on complex tasks like *Counting*, where object-level reasoning is critical. Additional baseline comparisons in Sec. C.1 further confirm the competitiveness of SCENECOT.

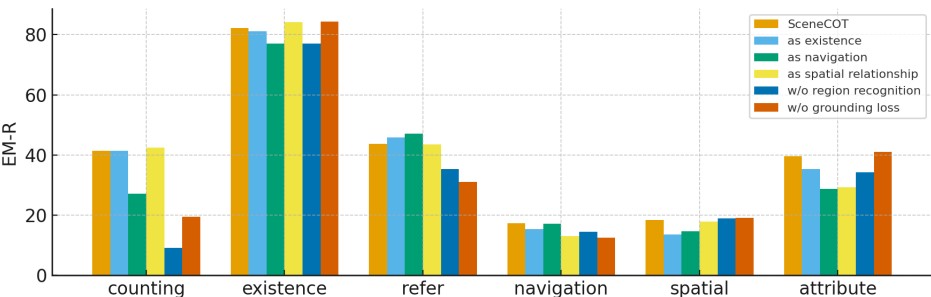

Figure 5: **Ablation study**. We ablate three key factors: (1) question type recognition, (2) region recognition, and (3) grounding loss. Results show that removing any of these components degrades performance, highlighting their importance for robust step-by-step reasoning.

**SCENECOT achieves strong grounding-QA coherence**. On Beacon3D, SCENECOT achieves the highest Good Coherence (34.7), substantially ahead of all baselines. This demonstrates that our framework most reliably combines correct grounding with correct QA, rather than succeeding on one dimension while failing on the other. By contrast, LEO reaches a QA (Obj.) score of 7.8 but suffers from an extremely low GC of 1.6, highlighting a large mismatch between answering correctly and grounding correctly. Similarly, methods like SceneVerse (GC: 20.4, QA (Obj.): 6.6) and Chat-Scene (GC: 19.5, QA (Obj.): 7.8) show moderate QA performance but struggle to align it with consistent grounding. In addition to its advantage in GC, SCENECOT also delivers the highest QA (Obj.) score (23.2), a large margin over all other methods. This reflects not only stronger per-object QA ability but also a tighter alignment between answers and the objects they refer to. Together, these results confirm that explicitly enforcing grounding before reasoning enables SCENECOT to achieve both accurate QA and high grounding-QA coherence, setting it apart from prior baselines.

### 5.3 ABLATION STUDY AND ANALYSES

**Question type recognition is essential.** As illustrated in Fig. 2, SCENECOT first identifies the question type and builds the reasoning chain accordingly. For example, the symbolic engine outputs polar coordinates for *Navigation* and 3D bounding boxes for *Spatial Relationship*. To assess its role, we conduct an ablation where all questions are forced into the same type. As shown in Fig. 5, treating all questions "as existence" restricts the model to object probabilities only, leading to clear performance degradation. Likewise, forcing all questions to *Spatial Relationship* significantly reduces performance on *Navigation*, where polar coordinates are more suitable. These results demonstrate that recognizing the correct question type is crucial for constructing grounded reasoning chains.

**Region recognition is crucial, particularly for object-centric reasoning**. To enhance grounding in Situated Reasoning, SCENECOT explicitly filters out irrelevant objects, narrowing the scope of candidates to task-relevant regions. Removing this component and providing all scene objects results in a substantial performance drop, especially on *Counting*, *Refer*, and *Attribute* (see Fig. 5). This confirms that region recognition not only reduces noise but also strengthens grounding-QA coherence by aligning reasoning with localized evidence.

**Grounding loss plays a key role**. In SCENECOT, the grounding module is trained with an additional loss term based on PQ3D, encouraging accurate object grounding during learning. Ablating this grounding loss (see Eq. (1)) yields a noticeable performance decline on MSQA, particularly for *Counting*, *Refer*, and *Navigation* (see Fig. 5). Interestingly, performance on *Existence* is less affected, likely because semantic labels alone suffice for this type of question. Overall, these results highlight the grounding loss as an important factor for maintaining step-by-step grounding consistency and improving reasoning accuracy.

**Exploring the QA upper bound with better grounding information**. In MSQA, predictions are limited by noisy semantic labels, masks, and object probabilities. To measure their impact, we test oracle settings that remove: ❶ Semantic errors (SE): inaccurate labels and masks; ❷ Grounding errors (GE): incorrect object probabilities. Removing SE improves performance, especially on *Counting*. Removing both SE and GE pushes results close to the oracle bound, with *Counting/Existence* near 100 and *Navigation/Refer* reaching 87.2/84.9. These gains confirm that accurate grounding and

Table 3: **Experimental results on oracle data**. In our main results, we utilize Mask3D to provide object masks and semantic labels. In this table, we explore the upper boundary in two aspects: 1) perfect object masks and semantic labels, but still based on the predicted object probabilities. 2) Oracle ground-truth text-based thought. In the table, "SE" indicates semantic error, "GE" indicates grounding error.

| Error sources | Count. | Exist. | Attri. | Spatial. | Refer | Navi. | Others | Overall |
|---|---|---|---|---|---|---|---|---|
| SE + GE | 47.9 | 82.1 | 49.6 | 49.8 | 31.9 | 51.6 | 80.3 | 55.6 |
| GE | 73.3 | 86.5 | 63.3 | 49.9 | 67.2 | 55.4 | 81.8 | 64.9 |
| – | 98.8 | 100.0 | 60.1 | 55.7 | 84.9 | 87.2 | 86.8 | 78.1 |

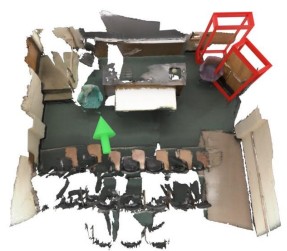
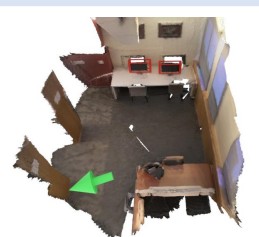

Figure 6: **Visualization of qualitative examples of SCENECOT.** We select two indicating sub-tasks *Counting* and *Navigation* to illustrate the reasoning traces of SCENECOT. Left:SCENECOT correctly constructs the visual clue and reasons the correct answer. Middle: SCENECOT correctly answers the question based on the accurate relative location. Right: Even though the visual clue exactly matches the correct entity, the model summarizes to the wrong answer owing to the limited reasoning capability.

semantics directly translate into better answers, demonstrating strong grounding-QA coherence in our framework. *Spatial Relationship* remains the hardest task, as it requires mapping coordinates to natural descriptions. Overall, the analysis shows that improving label, mask, and grounding quality is key to advancing 3D reasoning.

## 5.4 ADDITIONAL EVALUATION OF GROUNDING PERFORMANCE

To more thoroughly evaluate the grounded aspect of our 3D-CoTs reasoning, we consider both traditional grounding benchmarks and QA-driven grounding benchmarks. As shown in Tab. 4, we incorporate multiple benchmarks to comprehensively evaluate performance in both **grounding-only settings** (Nr3D, Beacon3D) and **QA-driven grounding tasks** (MSQA, SQA3D, ScanQA). For the latter, the evaluation measures the model's ability to localize the specific object instances required to answer a question (e.g., grounding only the specific chairs to the agent's right). We provide the details of the benchmarks in Sec. C.3.

The results show that SCENECOT achieves strong and balanced grounding performance across all benchmarks. It performs competitively on Nr3D and Beacon3D, and it significantly surpasses prior methods on MSQA, SQA3D, and ScanQA grounding. Notably, our model is not fine-tuned on SQA3D or ScanQA; thus, the strong **zero-shot grounding performance** highlights the effectiveness and generalizability of our approach. These high-quality grounding outputs provide clear and reliable visual cues that support the downstream reasoning chains.

Table 4: **Evaluation of grounding tasks.** "B3D" denotes "Beacon3D" in the table.

| Method | Nr3D | B3D | MSQA | SQA3D | ScanQA |
|---|---|---|---|---|---|
| | Top-1 | Top-1 | F1@50 | F1@50 | F1@50 |
| Chat-Scene | 39 | 62.7 | 15.9 | 25.9 | 35.9 |
| PQ3D | **66** | **76** | 10.6 | 10.8 | 19.2 |
| SceneCOT | 57.7 | 67.8 | **52.1** | **51.6** | **40.8** |

In contrast, while baseline models like PQ3D demonstrate strong results on traditional benchmarks like Nr3D, they perform poorly on QA-driven grounding tasks. This reveals a limitation in their ability to follow complex instructions in reasoning scenarios. Chat-Scene shows relatively better grounding on SQA3D and ScanQA but still lags behind SCENECOT across the full spectrum of tasks. Overall, this evaluation demonstrates that our framework is built upon robust grounding capabilities for both spatial localization and reasoning-oriented tasks.

## 5.5 REASONING CHAIN VISUALIZATION

Finally, we present a case study in Fig. 6 to illustrate the strengths and limitations of our framework. In the first example, the model grounds the doors at the agent's 2 o'clock and constructs the corresponding visual clue `obj_prob`. Although the predicted labels differ slightly from the ground-truth "door," the accurate probabilities and semantic similarity allow the model to generate the correct answer. In the second example, the model identifies all monitors and uses the symbolic engine to compute their relative positions with 2D polar coordinates. By integrating spatial information with object probabilities, the model infers the answer "4 o'clock." These two cases demonstrate interpretable reasoning chains for challenging tasks, making error sources easier to diagnose. In the third example, however, the model fails in the final reasoning step despite an accurate visual clue, echoing the upper-bound analysis in Tab. 3 where *Navigation* performance peaks at 87.2. This highlights a gap due to limited reasoning capability in the base model, which could be improved as foundation models like multimodal LLM advance. Overall, these visualizations showcase both the progress and remaining challenges of SCENECOT, pointing toward more robust and interpretable 3D reasoning frameworks.

## 6 CONCLUSION

We presented SCENECOT, a framework for step-by-step grounded reasoning in 3D scenes. Unlike prior approaches that treat 3D question answering as a single-step task, SCENECOT decomposes reasoning into task recognition, region localization, entity grounding, and grounded reasoning, closely reflecting the human problem-solving process. To enable this, we built SCENECOT-185K, the first large-scale dataset of grounded Chain-of-Thought reasoning traces with rich stepwise annotations linking language and 3D context. Experiments show that SCENECOT achieves strong performance on both situated reasoning benchmarks and the Beacon3D benchmark, where it significantly improves grounding-QA coherence compared to object-centric baselines. Beyond accuracy, our method generates interpretable reasoning traces that make the decision process transparent. These results demonstrate that step-by-step grounded reasoning is both effective and necessary for robust 3D scene understanding, laying a foundation for advancing multimodal LLMs toward human-like reasoning in real-world 3D environments.

## REPRODUCIBILITY STATEMENT

In this paper, we propose a new reasoning framework together with a large-scale dataset. We present the implementation details of the model design and training in both the main paper and the appendix. In Sec. 3.1, we define the different types of thoughts based on question categories. In Sec. 3.2, we describe how the modular components are trained and the pre-trained models we adopt. Hyperparameters used for training and evaluation are provided in Sec. A.5. Additional details include the definition of the symbolic engine in Sec. A.2, the visual clue construction algorithm in Sec. A.3, and the design of the 3D grounding module in Sec. A.3.

For data generation, we outline the full pipeline, including metadata collection in Sec. 4.1 and reasoning trace generation in Sec. 4.2. To facilitate reproducibility, we also provide the detailed system prompt used in Sec. B.2.

## ETHICS STATEMENT

This work adheres to the ICLR Code of Ethics. Our study focuses on improving 3D scene reasoning through step-by-step grounded reasoning. We use only publicly available 3D vision-language datasets, none of which contain personal or sensitive information. No human subjects were involved in this research.

Our method inherits biases and limitations from pre-trained models and datasets, such as potential semantic or linguistic biases in object labels and answers. While our framework improves interpretability and grounding-QA coherence, it is not intended for direct deployment in safety-critical domains (*e.g.*, autonomous navigation or surveillance) without further safeguards. Code and processed data will be released to support transparency and reproducibility.

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

# A    SCENECOT FRAMEWORK

## A.1    DATA EXAMPLES OF REASONING TRACES

We provide typical examples of COT templates of some representative sub-tasks, including the integral special tokens and textual context.

---

Question: You are climbing the shelf. There is a black and white monitor and a white and black monitor at your 5 o'clock. There is a brown door at your 2 o'clock. How many pillows are on your right in the middle distance?
Response:
<think_type>This is a counting question, so I need to ground the corresponding objects to answer it.</think_type>
<grd_rgn>
<think_rgn>Now I need to list all the objects on my right in the middle distance</think_rgn>
<think_grd>Ground the object: The pillows on my right in the middle distance. You should find all the possible objects.</think_grd>
[OBJ]
<think_task>Now I need to list all the potential objects and the probability.</think_task>
<list_obj_prob>
<obj_prob>pillow: 0.74 pillow: 0.78 cabinet: 0.31 legs: 0.01 cabinet: 0.29 pillow: 0.14 pillow: 0.40 file cabinet: 0.31 wall: 0.04 whiteboard: 0.45</obj_prob>
<think_sum>Now answer the question based on the object probabilities.</think_sum>
<answer>two</answer>

---

Figure 7: **A COT example of *Counting*.**

---

Question: You are painting the door. There is a brown trash can in your 9 o'clock direction. There is a turquoise table with a smooth texture and a four-legged structure in your 6 o'clock direction. Is there a sink in the room?
Response:
<think_type>This is an existence question, so I need to ground the corresponding objects to answer it.</think_type>
<grd_rgn>
<think_rgn>Now I need to list all the objects in the scene.</think_rgn>
<think_grd>Ground the object: The sinks in the room.</think_grd>
[OBJ]
<think_task>Now I need to list all the potential objects and the probability.</think_task>
<list_obj_prob>
<obj_prob>sink: 0.95 traffic cone: 0.24 folded table: 0.06 ladder: 0.20 trash can: 0.45 vacuum cleaner: 0.35 ladder: 0.19 ledge: 0.14 table: 0.43 floor: 0.23 recycling bin: 0.13 recycling bin: 0.05 rack: 0.11 folded table: 0.08 boxes: 0.46 trash can: 0.40 trash can: 0.36 folded table: 0.16 wall: 0.28 door: 0.48 doorframe: 0.37 wall: 0.28 sink: 0.04 table: 0.15 wall: 0.18 wall: 0.38 trash can: 0.26 wall: 0.32 wall: 0.18 wall: 0.04</obj_prob>
<think_sum>Now answer the question based on the object probabilities.</think_sum>
<answer>yes</answer>

---

Figure 8: **A COT example of *Existence*.**

Question: There is a window in your 7 o'clock direction. There is a black monitor with a smooth texture in your 3 o'clock direction. What object is brown and used for seating at your 6 o'clock?
Response:
<think_type>This is a refer question, so I need to ground the corresponding objects to answer it.</think_type>
<grd_rgn>
<think_rgn>Now I need to list all the objects at my 6 o'clock.</think_rgn>
<think_grd>Ground the object: The brown object used for seating at your 6 o'clock.</think_grd>
[OBJ]
<think_task>Now I need to list all the potential objects and the probability.</think_task>
<list_obj_prob>
<obj_prob>chair: 0.98 wall: 0.30 floor: 0.33 chair: 0.48 chair: 0.44 table: 0.10 window: 0.26 backpack: 0.40</obj_prob>
<think_sum>Now answer the question based on the object probabilities.</think_sum>
<answer>the chair</answer>

Figure 9: **A COT example of *Refer*.**

Question: You are replacing the trash bag in the trash can. In front of you, there's a beige wooden door. To your left, at a middle distance, there's a dark rectangular suitcase with a hard shell. What is the state of the door on your right?
Response:
<think_type>This is an attribute question, so I need to ground the corresponding objects to answer it.</think_type>
<grd_rgn>
<think_rgn>Now I need to list all the objects in the sub-region related to the question.</think_rgn>
<think_grd>Ground the object: The door on my right.</think_grd>
[OBJ]
<think_task>Now I need to retrieve the image of the target object.</think_task>
<highlight_obj>
<think_sum>Now answer the question based on the object image.</think_sum>
<answer>closed</answer>

Figure 10: **A COT example of *Attribute*.**

## A.2    DEFINITION OF THE SYMBOLIC ENGINE

Our symbolic engine serves two primary functions:

1. **Spatial Region Recognition**: It partitions the environment into sub-regions based on the agent's location and orientation.

2. **Visual Clue Construction**: It generates textual visual clues by integrating object probabilities, semantic labels, and spatial locations.

### A.2.1    SPATIAL REGION RECOGNITION

In MSQA, there are two types of directional reference policies: 1) *Cardinal-Relative Directions*, such as left, right, front, and behind. 2) *Clock-based directions*, such as 1 to 12 o'clock positions. Here we provide some code fragments of our implementation.
1. Calculate the cardinal-relative sub-regions. This function is used to calculate the object lists of four directions. Since MSQA also requires the distance information in some scenarios, we also develop the function to support this feature.
2. Calculate the clock-based sub-regions. This function is used to calculate the clockwise information for each object.

Question: You are sitting at the desk. Behind you, there's a burgundy backpack. From where You are, how do You get to the door? Please provide a direction (behind, in front of, left, right) and the distance information (far/middle/near).
Response:
<think_type>This is a navigation question, so I need to ground the corresponding objects first.</think_type>
<grd_rgn>
<think_rgn>Now I need to list all the objects in the scene.</think_rgn>
<think_grd>Ground the object: The door.</think_grd>
[OBJ]
<think_task>Now I need to list all the locations(polar coordinate: angle, distance(/m)) and probabilities of the potential objects.</think_task>
<list_obj_loc_plr_prob>
<obj_loc_plr_prob>door: -86.1, 1.9; prob: 0.79 wall: -117.6, 2.7; prob: 0.01 window: 154.2, 2.0; prob: 0.25 fan: 152.1, 1.6; prob: 0.37 clothes: 96.4, 3.4; prob: 0.41</obj_loc_plr_prob>
<think_sum>Now answer the question based on the object probabilities.</think_sum>
<answer>turn right and walk to the middle distance.</answer>

Figure 11: **A COT example of *Navigation*.**

Question: You are hanging the picture. There is a blue tissue box at your 10 o'clock direction, with a smooth cardboard texture, dispensing tissues. A cluttered dark table is at your 7 o'clock direction. Where is the book located in relation to the desk?
Response:
<think_type>This is a spatial relationship question, so I need to ground the corresponding objects first.</think_type>
<grd_rgn>
<think_rgn>Now I need to list all the objects in the scene.</think_rgn>
<think_grd>Ground the object: The book and the desk.</think_grd>
[OBJ]
<think_task>Now I need to list all the locations and probabilities of the potential objects.</think_task>
<list_obj_loc_prob>
<obj_loc_plr_prob>book: -1.6,4.8,0.8,0.3,0.3,0.1; prob: 0.89 desk: -0.9,5.0,0.5,1.7,0.8,0.9; prob: 0.59 yoga mat: -2.5,4.6,0.7,0.6,0.2,0.2; prob: 0.46 shelf: -1.2,1.3,1.0,0.6,0.9,2.1; prob: 0.05 sink: 3.0,2.6,0.8,0.4,0.5,0.4; prob: 0.06 bag: -1.5,2.8,0.1,0.6,0.3,0.3; prob: 0.14 book: -1.6,4.8,0.8,0.3,0.3,0.1; prob: 0.17 kitchen island: 2.7,2.6,0.4,1.2,1.8,1.0; prob: 0.36 refrigerator: 2.8,4.6,0.9,1.0,0.4,1.7; prob: 0.11 lamp base: -1.4,2.2,0.0,0.2,0.2,0.1; prob: 0.33</obj_loc_plr_prob>
<think_sum>Now answer the question based on the object probabilities.</think_sum>
<answer>the book is placed within the area of the desk.</answer>

Figure 12: **A COT example of *Spatial Relationship*.**

3. We parse the text between `<think_rgn>` and `</think_rgn>` to obtain the user's directional instruction, then set `query_type` by matching it to known phrases like "on my left" or "at my 10 o'clock".

### A.2.2 VISUAL CLUE CONSTRUCTION

In the Grounded Reasoning step, the model generates the final answer by integrating both semantic information and a visual clue derived from object probabilities. We outline the algorithm for constructing this visual clue during the reasoning process. Given a prefix sequence $\mathcal{P}$ (i.e., the input_ids of the text prompt), the algorithm returns a modified sequence that incorporates relevant visual clues based on object probabilities, spatial locations, and semantic labels. This enhanced sequence is then fed back into the LLM to generate the final response. The component functions—build_obj_prob, build_obj_loc_prob, and build_obj_loc_plr_prob—are illustrated in Fig. 16 and Fig. 3. Additionally,

Question: What color is the printer on top of the nightstand?
Response:
<think_type>This is a appearance (grounded qa) question, so I need to ground the corresponding objects to answer it.</think_type>
<grd_rgn>
<think_rgn>Now I need to list all the objects in the scene.</think_rgn>
<think_grd>Ground the object: The printer on top of the nightstand.</think_grd>
[OBJ]
<think_task>Now I need to retrieve the image of the target object.</think_task>
<highlight_obj>
<think_sum>Now answer the question based on the object image.</think_sum>
<answer>Black</answer>

Figure 13: **A COT example of *GQA3D*.**

```python
# calculate the sub-graphs for Cardinal-Relative Directions

face_pt = np.array([np.cos(direction), np.sin(direction)])

stand_pt = np.array([stand_on_loc[0], stand_on_loc[1]])
pcds = np.array(pcds)
pcd_2d = pcds[:, :2]

pcd_2d = pcd_2d - stand_pt
pcd_2d_norm = np.linalg.norm(pcd_2d, axis=1)
pcd_2d_norm[pcd_2d_norm == 0] = 1
pcd_2d_norm = np.expand_dims(pcd_2d_norm, axis=1)
pcd_2d = pcd_2d / pcd_2d_norm

### cal the angle between face_pt and pcd_2d
sum_all = np.dot(pcd_2d, face_pt)
sum_all[sum_all >= 1] = 1
sum_all[sum_all <= -1] = -1
angle = np.arccos(sum_all)
angle = angle / np.pi * 180

### cal the cross value between face_pt and pcd_2d
cross = np.cross(face_pt, pcd_2d)

front_mask = (angle < 30)
back_mask = (angle > 150)
left_mask = (cross > 0) & (angle > 30) & (angle < 150)
right_mask = (cross < 0) & (angle > 30) & (angle < 150)

front_list = get_inst_id(inst_label, front_mask)
back_list = get_inst_id(inst_label, back_mask)
left_list = get_inst_id(inst_label, left_mask)
right_list = get_inst_id(inst_label, right_mask)
```

Fig. 14 provides a visual comparison between the 2D polar coordinate system and the 3D spatial coordinate system used in our framework.

### A.3 IMPLEMENTATION DETAILS OF THE 3D VISUAL GROUNDING MODULE

We illustrate the implementation of 3D Visual Grounding Module in Fig. 15. In the framework, the object feature is extracted by PQ3D, while the text embedding is extracted by the embedding tokenizer based on the grounding text. We train PQ3D based on the grounding data in Jia et al. (2024), including the scenes in ScanNet, 3RScan, and MultiScan.

```python
def cal_clock_wise_direction(stand_on_pt, face_pt, inst_pcd):
    inst_pcd = inst_pcd[:,:2]
    # inst_loc = (inst_pcd.max(0) + inst_pcd.min(0)) / 2
    inst_loc = inst_pcd.mean(0)
    obj_dir = inst_loc - stand_on_pt
    if np.abs(obj_dir).sum() < 1e-6:
        return None, None

    obj_dir = obj_dir / np.linalg.norm(obj_dir)
    face_pt = face_pt / np.linalg.norm(face_pt)

    ### cal the angle between obj_dir and face_dir
    angle = np.dot(obj_dir, face_pt)
    angle = np.clip(angle, -1, 1)
    angle = np.arccos(angle)
    direct = np.cross(face_pt, obj_dir)
    if direct > 0:
        angle = 2 * np.pi - angle
    angle = angle / np.pi * 180

    clockwise_direction = round(angle / 30) % 12
    clockwise_direction = 12 if clockwise_direction == 0 else
        clockwise_direction
    clockwise_direction = int(clockwise_direction)

    return clockwise_direction, angle
```

```python
# query: the object list of: cardinal direction-based and clock-based
# query type: four direction or clockwise or whole_scene
if query_type == 'cardinal' or query_type == 'clockwise':
    parse_words = ['left', 'right', 'front', 'back', 'behind'] if
        query_type == 'cardinal' else [f"{i} o'clock" for i in range(12)]
    direction = parse_direction_distance(question, parse_words)
    direction = direction if direction != 'behind' else 'back'
    distance = parse_direction_distance(question, ['far', 'middle', 'near
        '])
    if direction:
        obj_list, label_ids = get_obj_list(query, query_type, direction,
            distance)
        count_dict = Counter(obj_list)
    else:
        # no direction found, collect objects in all directions
        obj_list, label_ids = [], []
        distance = parse_direction_distance(question, ['far', 'middle', '
            near'])
        for direction in query.keys():
            obj_list_temp, label_ids_temp = get_obj_list(query,
                query_type, direction, distance)
            obj_list += obj_list_temp
            label_ids += label_ids_temp
        count_dict = Counter(obj_list)
else:
    obj_list, label_ids = get_obj_list(query, query_type, '', '')
    count_dict = Counter(obj_list)
```

### A.4 INFERENCE

Recently, several state-of-the-art industrial LLMs—such as DeepSeek-V3 Liu et al. (2024)—have adopted the Mixture-of-Experts (MoE) technique as a key performance-enhancing strategy. Inspired by this insight, we implement a minimal routing mechanism during the inference stage. Empirically, we observe that the model's performance on several sub-tasks such as *Existence* and *Attribute* sub-

---

**Algorithm 1:** Visual Clue Construction for Grounded Reasoning

---

**Require :** object probabilities $\mathcal{P} \in R^{N \times 1}$, prefix sequence $S \in R^{N \times M}$,
        object locations and sizes $\mathcal{O}_L \in R^{N \times 6}$, object semantic labels $\mathcal{O}_{SL}(N \times 1)$,
        object images $\mathcal{O}_I \in R^{N \times 3 \times H \times W}$, maximum object number $K$

**if** `LIST_OBJ_PROB_TOKEN_IDX` *in* $S$ **then**
    `obj_prob_content` ← build_obj_prob($\mathcal{P}, \mathcal{O}_L, \mathcal{O}_{SL}, K$);
    `index` ← Index($S$, `LIST_OBJ_PROB_TOKEN_IDX`);
    `new_sequence` ← cat($S$`[:index+1]`, get_text_embeddings(`obj_prob_content`));

**else if** `LIST_OBJ_PROB_LOC_TOKEN_IDX` *in* $S$ **then**
    `obj_prob_loc_content` ← build_obj_loc_prob($\mathcal{P}, \mathcal{O}_L, \mathcal{O}_{SL}, K$);
    `index` ← Index($S$, `LIST_OBJ_PROB_LOC_TOKEN_IDX`);
    `new_sequence` ← cat($S$`[:index+1]`,
     get_text_embeddings(`obj_prob_loc_content`));

**else if** `LIST_OBJ_PROB_LOC_PLR_TOKEN_IDX` *in* $S$ **then**
    `obj_prob_loc_plr_content` ← build_obj_loc_plr_prob($\mathcal{P}, \mathcal{O}_L, \mathcal{O}_{SL}, K$);
    `index` ← Index($S$, `LIST_OBJ_PROB_LOC_PLR_TOKEN_IDX`);
    `new_sequence` ← cat($S$`[:index+1]`,
     get_text_embeddings(`obj_prob_loc_plr_content`));

**else if** `HIGHLIGHT_OBJ_TOKEN_IDX` *in* $S$ **then**
    `top_k_indices` ← TopK($\mathcal{P}$);
    `img` ← $\mathcal{O}_I$`[top_k_indices[0]]`;
    `img_tokens` ← Projector(Img_encoder(`img`));
    `index` ← Index($S$, `HIGHLIGHT_OBJ_TOKEN_IDX`);
    `new_sequence` ← cat($S$`[:index+1]`,
     get_text_embeddings(`IMG_START_TOKEN_IDX`), `img_tokens`,
     get_text_embeddings(`IMG_END_TOKEN_IDX`));

**return** `new_sequence`

---

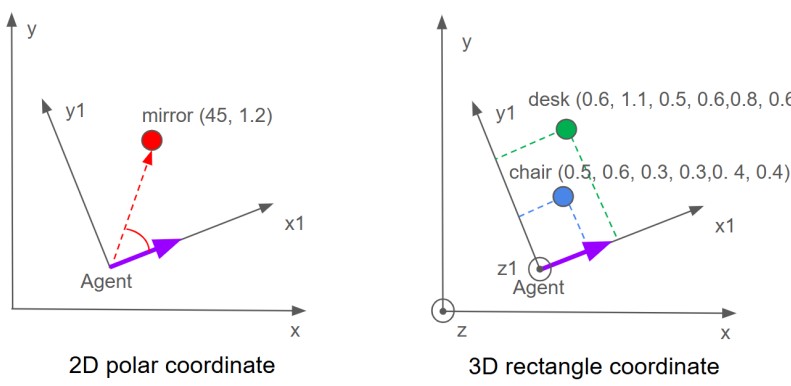

Figure 14: **Coordinate system comparison**. We provide a diagram of the two types of coordinate systems.

tasks tends to degrade over time, while performance on other tasks improves. Motivated by this training dynamic, we introduce a simple two-expert selection strategy.

Specifically, the model with the highest overall validation performance is designated as **Expert-1**, while the model with the best validation performance on partial sub-tasks is designated as **Expert-2**. During inference, if the predicted question type comes from the ones that have a gradual decreasing trend in validation performance, the input prefix sequence is routed to Expert-2; otherwise, it is routed to Expert-1. This strategy feasibly utilizes the result of task recognition in the first reasoning step. This strategy is only applied for MSQA in evaluation. Balancing the training dynamics for different sub-tasks should be an important direction for future work.

As we apply `LoRA` to the LLM during training, this inference strategy introduces only an additional 330M parameters relative to a single base model, representing a reasonable trade-off between performance and deployment cost.

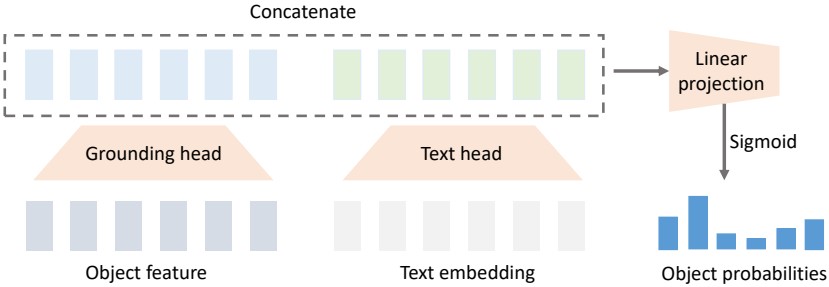

Figure 15: **3D Visual Grounding Module Design**.

Table 5: **Hyperparameters for training.**

| Hyperparameter | Value |
| --- | --- |
| Optimizer | AdamW |
| Weight Decay | 0.05 |
| betas | [0.9, 0.999] |
| Learning Rate | $3 \times 10^{-5}$ |
| Warmup Steps | 400 |
| Type of GPUs | NVIDIA A100 |
| Number of GPUs | 4 |
| Accumulate Gradient Batches | 5 |
| Batch Size/GPU (total) | 2 (80) |
| gradient norm | 5.0 |
| epochs | 5 |

Table 6: Hyperparameters for inference.

| Hyperparameter | Value |
| --- | --- |
| Number of beams | 5 |
| maximum output length | 256 |
| repetition penalty | 3.0 |
| length penalty | 1.0 |

### A.5 HYPERPARAMETERS FOR MODEL TRAINING AND INFERENCE

We train the base LLM of SCENECOT in a single stage, initializing from the pretrained weights of LLaVA-1.5. The detailed hyperparameter settings are provided in Tab. 5 and Tab. 6.

## B SCENECOT-185K DATASET

### B.1 DATA GENERATION PIPELINE OF SCENECOT-185K

Our data generation pipeline is shared between Object-Centric Reasoning and Situated Reasoning, with Object-Centric Reasoning formulated as an *Attribute* sub-task under the broader Situated Reasoning framework. The overall process is illustrated in Fig. 16.

### B.2 DATA GENERATION DETAILS AND EXAMPLES OF GQA3D

To enable the training of Object-Centric Reasoning in non-situated scenarios, we incorporate GQA3D as a key component of the overall dataset. The primary distinction from Situated Reasoning lies in the nature of the grounding text, which describes the target object from a global perspective rather than an egocentric view. Additionally, this task does not involve multi-object grounding or reasoning. We use GPT-4o to generate diverse question-answer pairs based on a given object image and its corresponding grounding text. We provide the system prompt in Fig. 17.

### B.3 DATA QUALITY

We conducted a manual quality check to verify the data quality. First, we manually checked the object IDs in MSQA. Note that all object IDs outside the scene were filtered out in our dataset. Since MSQA is based on situated scene graphs, all objects in QA pairs correspond to specific sub-regions,

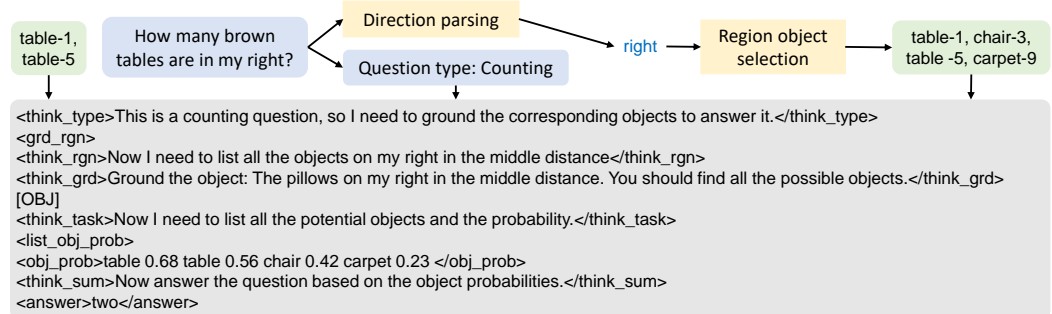

Figure 16: **Data Generation Procedure of *Counting*.**

You are an expert visual question-answer pair generator.
You will receive: An image showing part of a scene. An object list that names all the objects supposed to be present in the scene. Several grounding texts of the target object, which refers to a specific object that appears in the image. Your task is to generate multiple high-quality QA (Question-Answer) pairs based on the image and the object list, focusing on the target object. The QA pairs should be diverse and cover five categories: existence, appearance, geometry, spatial relationship, and class.
Follow these detailed guidelines:
1. Existence Questions Generate both yes and no questions. For questions with the answer "Yes": refer to objects that are adjacent to the target object in the image. For questions with the answer "No": refer to objects listed in the object list but absent in the image.
2. Spatial Relationship Questions Ask about the spatial relations involving the target object (e.g., "next to," "above," "below," "in front of," "close to," "adjacent to" etc. Please do not use the directional words like 'left' or 'right'). Questions should describe how the target object is positioned relative to nearby objects.
3. Class Questions Ask about the category or type of objects adjacent to the target object. Examples: "What object is next to the target object?" or "What is under the target object?"
4. Geometry Questions Ask about the shape and size comparison of the target object. Examples: "What is the shape of the target object?" or "Which is taller, the target object or another nearby object?"
5. Appearance Questions Ask about visible attributes like color, texture, or material of the target object.
Important Notes:
Base all questions and answers on the visual content of the image and the object list. For questions with the answer 'Yes', make sure they are visually supported by the image. Ensure that every QA pair is grounded and accurate to the given visual context.
Always specify the question type (existence, appearance, geometry, spatial relationship, class) when you output a QA pair. Use the target object description naturally inside the questions to avoid ambiguity. Do not ask the questions that have been mentioned in the grounding text. You can choose the objects in the object list. While all the answers of the questions are based on the given image.
Output format: Q: [question] A: [answer] type: [type]

Figure 17: **System Prompt for QA Pairs Generation in GQA3D.**

as ensured by the MSQA paper. We randomly sampled 50 examples each from counting, spatial relationship, refer, attribute, and navigation tasks, totaling 250 examples. Annotators were given the question, answer, and object IDs to verify if the IDs align with the QA pairs. For example, if the question is "How many chairs are on my left?" with the answer "three," but the target object IDs are ["chair-1", "chair-3"], then the annotation is incorrect. The results are shown in Tab. 8. The results demonstrate the reliability of our annotation process.
Next, we verify the quality of the generated QA pairs in GQA3D. GQA3D uses Nr3D as its data source to create QA pairs with reasoning traces. The grounding text is mostly human-annotated, ensuring data quality. We generated new QA pairs based on object images using GPT-4V. To ensure

Table 7: **Metadata Examples of GQA3D**. GQA3D constructs QA pairs using the grounding text from Nr3D. The QA pairs are generated from the image of a target object.

| Object Image | Grounding Text | QA pairs |
|---|---|---|
|  | the tallest white box, the tall box to the left of the 2 boxes sitting on top of each other, the tall white box that is not stacked | Q: What material is 'the tallest white box' made of? A: Cardboard type: appearance Q: Is 'the tall box to the left of the 2 boxes sitting on top of each other' the same color as the door? A: No type: existence Q: Is there a fan next to 'the tall white box that is not stacked'? A: No type: existence |

Table 8: **Data quality of MSQA**. We check the accuracy of target object IDs.

|  | counting | spatial | refer | attribute | navigation |
|---|---|---|---|---|---|
| # sampled instances | 50 | 50 | 50 | 50 | 50 |
| accuracy | 92% | 100% | 98% | 100% | 100% |

Table 9: **Data quality of GQA3D**. We sample some data instances and check the answer correctness.

| # sampled instances | # sampled objects | accuracy |
|---|---|---|
| 100 | 11 | 90% |

quality, we randomly sampled 100 QA pairs and manually verified their correctness, providing both the object images and QA pairs for review. The results are shown in Tab. 9. The results confirm the reliability of our data. The strong performance of Chat-Scene and SceneCOT further highlights the value of our dataset.

## C ADDITIONAL EXPERIMENTS AND ANALYSES

### C.1 ADDITIONAL BASELINE COMPARISON

In our main results, we focus on object-centric methods such as Chat-Scene, LEO, and MSR3D to clearly analyze both QA performance and grounding-QA coherence. To broaden the comparison, we also include LLaVA-3D as a strong voxel-based baseline. As shown in Tab. 10, SCENECOT achieves competitive performance on both Situated Reasoning and Object-Centric Reasoning. An interesting direction for future work is integrating SCENECOT with voxel-based approaches.

### C.2 ADDITIONAL EVALUATION AND BASELINE COMPARISON

We assess our model on SQA3D and ScanQA under zero-shot settings and compare it with Chat-Scene and LEO. Both Chat-Scene and LEO are fine-tuned on SceneCOT-QA; specifically, LEO is fine-tuned on the SceneCOT-QA dataset, while Chat-Scene is fine-tuned on the SceneCOT-grounded QA variant, where the model must first predict the target object IDs before generating the answer. The results are summarized in Table 11.

We evaluate both QA and grounding performance. The results indicate that our QA performance is comparable to Chat-Scene, while our grounding performance is substantially stronger. This reinforces our claim that although SceneCOT **does not rely on scene tokens**, it explicitly grounds objects before answering. Since answers must be inferred solely from grounded results, the model demonstrates strong grounding–QA coherence, whereas Chat-Scene shows weak coherence between its QA performance and grounding performance.

### C.3 DETAILS OF QA-DRIVEN GROUNDING BENCHMARKS

Table 10: **Additional baseline on MSQA and Beacon3D**.

| Methods | Data | Grounded? | MSQA | | | | | | | Beacon3D | |
| | | | *Count.* | *Exist.* | *Attr.* | *Spatial* | *Navi.* | *Others* | Overall | Case | Obj. |
|---------|------|-----------|---------|---------|---------|---------|---------|---------|---------|------|------|
| LEO | ours | ✗ | 29.3 | 87.5 | 55.1 | 46.3 | 45.6 | 81.4 | 52.9 | 52.5 | 12.7 |
| MSR3D | ours | ✗ | 32.7 | 87.5 | 53.7 | 44.3 | 51.5 | 72.3 | 52.8 | 51.4 | 11.9 |
| Chat-Scene | ours | ✗ | 37.4 | 92.0 | 49.0 | 47.0 | 58.3 | 83.7 | **56.6** | 53.6 | 14.0 |
| LLaVA-3D | ours | ✗ | 38.5 | 89.5 | 57.0 | 49.8 | 38.5 | 84.4 | 54.9 | **59.1** | 21.0 |
| SCENECOT | ours | ✓ | 47.7 | 82.1 | 49.6 | 47.2 | 51.6 | 80.3 | 55.6 | 58.9 | **23.2** |

Table 11: **Zero-shot QA and grounding performance on SQA3D and ScanQA.**

| Method | SQA3D (EM-R) | SQA3D-G (F1@50) | ScanQA (EM-R) | ScanQA-G (F1@50) |
|--------|--------------|-----------------|---------------|------------------|
| Chat-Scene | 36.0 | 3.4[*] | **24.5** | 4.1[*] |
| LEO | 36.7 | – | 20.04 | – |
| SceneCOT | **39.7** | **51.6** | 21.0 | **40.8** |

[*] **Note on Chat-Scene:** Despite following official instructions and using 8-GPU reproduction attempts, grounding performance remained low. Reports (e.g., GitHub Issue #38) suggest the model is highly sensitive to training settings.

In Sec. C.2, we evaluate the grounding performance of the models on ScanQA and SQA3D. For ScanQA, we utilize the object instance IDs provided by the official annotations. For SQA3D, we built an agent workflow and chose the rules defined in Sec. A.2 to extract the question-answer related object IDs. We denote the corresponding data as "ScanQA-G" and "SQA3D-G". To guarantee the data quality, we conducted manual quality verification. The results is shown in Tab. 12. The result reveals the reliability of SQA3D-G.

Table 12: **Data quality of SQA3D.**

| # Instance | Accuracy |
|------------|----------|
| 33 | 93.3% |

## C.4 ADDITIONAL REASONING CHAIN VISUALIZATION RESULTS

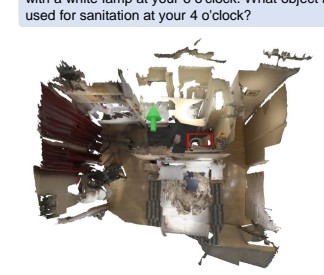
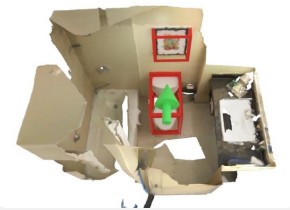
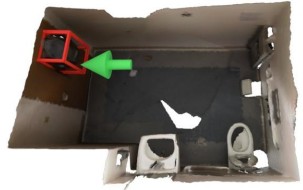

Figure 18: **Additional Reasoning Chain Visualization**. We provide more visualization results on *Refer*(Left), *Spatial Relationship*(Middle), and *Existence*(Right).

We present additional visualizations of the reasoning chains across multiple sub-tasks to provide a deeper understanding of our reasoning mechanism. In the first example, the model correctly

identifies the target object based on the semantic grounding text ("The object used for sanitation"), and subsequently arrives at the correct answer by leveraging accurate visual cues. In the second example, the grounding module successfully locates the target objects ("The toilet and the picture"), which enables the model to reason effectively using object coordinates. However, in another case, the grounding module fails to identify the target object ("backpack"), resulting in an incorrect answer. We also include a video demonstration to intuitively showcase the entire workflow.

# D  FURTHER ANALYSIS AND EFFICIENCY

## D.1  INFERENCE LATENCY AND TRADE-OFFS

Our framework primarily focuses on enabling a **reliable and transparent reasoning process**, which requires incorporating all necessary contextual information during inference. This approach inevitably incurs higher inference time due to multiple module calls and the construction of detailed reasoning contexts. We provide a qualitative comparison of latency and token length in Table 13.

Table 13: **Comparison of reasoning-token length and inference latency.**

| Method | Object instance information? | Reasoning token length | Inference latency |
|---|---|---|---|
| LEO | No | 0 | 4.8s |
| LLaVA-3D | No | 0 | 0.2s |
| Chat-Scene | No | 0 | 0.5s |
| SceneCOT | Yes | 350–1500 | 10.4s |

The majority of the latency in SceneCOT (10.4s total) stems from LLM sequence generation, specifically Stage 1–3 generation (4.2s) and Stage 4 generation (3.8s), while visual clue construction remains relatively moderate at 2.4s. We believe this represents a **reasonable trade-off** between transparent, interpretable reasoning and inference speed.

## D.2  ACCURACY OF TASK AND REGION RECOGNITION

In our framework, errors from Task Recognition and Region Localization can be largely ignored due to their high accuracy. As shown in Table 14, both sub-tasks achieve extremely high performance, which we attribute to the strong commonsense priors inherent in large language models.

Table 14: **Accuracy of internal recognition modules.**

| Module | Accuracy (%) |
|---|---|
| Question type recognition | 99.4 |
| Region recognition | 100.0 |

For Region Localization, we employ a rule-based filtering mechanism. For instance, given an instruction like *"Now I need to ground the objects on my right,"* the system applies predefined rules to filter objects within the corresponding sub-region, ensuring accurate object selection aligned with the model's recognition results.

# E  LIMITATIONS

Though we propose a first step-by-step reasoning framework and have demonstrated its advantages on typical 3D scene reasoning tasks. There are also several limitations in our work.
First, our framework focuses on the tasks pre-defined in MSQA, which is limited in more complex scenarios such as embodied AI. For example, we do not consider the long-horizon tasks such as embodied task planning. Recently, SG3D Zhang et al. (2024b) proposes a new benchmark to evaluate 3D-VL models' capabilities of grounded task planning in embodied scenarios. We will consider extending our framework to this task in the future.
Second, SCENECOT-185K is built upon MSQA-ScanNet and Nr3D, which only contains the 3D scenes in ScanNet. Extending the dataset to more diverse real-world scenes is an important direction to unlock the real-world applications in the future.
Third, our thought design is still not perfect in partial sub-tasks. Even the we have demonstrated promising upper boundaries in some challenging sub-tasks, such as *Counting* and *Navigation*, our thought design still struggles with solving problems like *Spatial Relationship*. How to design better

3D-CoTs is another important direction to further increase the upper boundaries of the reasoning framework in 3D scenes. Besides, based on the recent practice in advanced reasoning LLMs Guo et al. (2025); OpenAI (2025), exploring more learning algorithms such as Reinforcement Learning may also lead to more surprising capabilities for complex 3D scene reasoning.

## F  BROADER IMPACT

Understanding and reasoning in 3D scenes is a cornerstone capability for building intelligent embodied agents that can operate safely, reliably, and effectively in the physical world. Our proposed approach, SCENECOT, introduces a structured, interpretable framework for 3D scene reasoning by incorporating a Chain-of-Thought (CoT) paradigm into the 3D vision-language (3DVL) domain. By explicitly modeling intermediate reasoning steps such as task recognition, region localization, entity grounding, and grounded reasoning, SCENECOT offers a significant leap toward human-level understanding in complex 3D environments.

The potential societal benefits of this work are substantial. It enables advancements in a wide range of real-world applications, including domestic robotics, assistive technologies for individuals with disabilities, autonomous navigation systems, and intelligent agents for virtual and augmented reality. The stepwise, interpretable nature of our method also enhances transparency and safety, which are essential for deploying AI systems in human-centered environments.

At the same time, we acknowledge that powerful embodied AI systems can be misused in ways that may compromise privacy or safety, particularly in surveillance or military contexts. To mitigate such risks, we encourage responsible research practices, including dataset transparency, open evaluation protocols, and active engagement with the broader community on the ethical deployment of such systems. Our work serves as a foundation for building grounded and generalizable reasoning agents, and we hope it will inspire future research that aligns technological advancements with human values.

## G  LLM USAGE STATEMENT

We use LLM in two ways: (1) to aid with polishing the writing, including improving grammar and coherence of the manuscript; and (2) to generate a portion of training data for our model. The LLM does not contribute to any significant part of this work.

