# OpenReview forum: "SceneCOT: Eliciting Grounded Chain-of-Thought Reasoning in 3D Scenes"
_ICLR.cc/2026/Conference — ICLR 2026 Poster_

### Official Review · Reviewer_rnc2 · 2025-10-30

**Soundness:** 3
**Presentation:** 2
**Contribution:** 2
**Rating:** 4
**Confidence:** 5

**Summary:**

This paper introduces grounded Chain-of-Thought reasoning into 3D large language models, decoupling complex reasoning tasks into simpler and more manageable problems, and constructing corresponding visual cues through multimodal expert modules. To achieve this, the authors focus on dataset curation. By training on the proposed dataset, the model attains state-of-the-art performance on several 3D understanding benchmarks.

**Strengths:**

1. The first to introduce CoT reasoning into 3D understanding.
2. Proposes a large-scale dataset to support the study.
3. The model achieves leading performance across multiple benchmarks.

**Weaknesses:**

1. **Limited evaluation:** The authors primarily evaluate on MSQA and Beacon3D, while several widely used benchmarks such as ScanQA and SQA-3D are not considered.
2. **Limited data sources:** The annotations mainly originate from Nr3D and MSQA. Incorporating larger and more diverse datasets and scenes, such as MMScan[3], could enhance generalization.
3. **Object-centric input:** The method mainly relies on object-centric input, which requires an additional segmentation model during inference and thus limits broader applicability. The ablation results show that performance is highly dependent on segmentation labels. Since object-centric input already provides a strong spatial prior for grounded reasoning, it remains unclear whether the proposed method can generalize to video-based 3D LLMs such as Video-3D LLM[1] or LLaVA-3D[2].

[1] https://arxiv.org/abs/2412.00493 (CVPR 25)
[2] https://arxiv.org/abs/2409.18125 (ICCV 25)
[3] https://arxiv.org/abs/2406.09401 (NIPS 24)

**Questions:**

1. Can grounded large-scale datasets like **3D-GRAND** be integrated into your pipeline?
2. What is the purpose of introducing probability during reasoning? Are there any ablation studies to justify its effect?
3. The **LEO** model adopts a similar object-centric input as the proposed method. Why does the performance on MSQA decrease after training on your dataset? The effectiveness of the proposed reasoning mechanism remains questionable.

---

> ### Author Response · Authors · 2025-11-28
> **Response to Reviewer rnc2 (Part 1)**
>
> We sincerely thank the reviewer again for the thoughtful feedback. We hope these clarifications address all concerns, and we welcome further questions or suggestions.
>
> ---
>
> ### **W1. Limited evaluation**
>
> We kindly thank the reviewer for this suggestion. To provide additional evaluation, we assess our model on SQA3D and ScanQA under zero-shot settings and compare it with Chat-Scene and LEO. Both Chat-Scene and LEO are fine-tuned on SceneCOT-QA: LEO is fine-tuned on the SceneCOT-QA dataset, while Chat-Scene is fine-tuned on the SceneCOT-grounded QA variant, where the model must first predict the target object IDs before generating the answer. The results are as follows:
>
> ||SQA3D(EM-R)|SQA3D-grounding(F1-score @ IoU50)|ScanQA(EM-R)|ScanQA-grounding(F1-score @ IoU50)|
> |---|---|---|---|---|
> |Chat-Scene|36.0|3.4*|**24.5**|4.1*|
> |LEO|36.7|--|20.04|--|
> |SceneCOT|**39.7**|**51.6**|21.0|**40.8**|
>
>
> We evaluate both QA and grounding performance. The results indicate that our QA performance is comparable to Chat-Scene, while our grounding performance is substantially stronger. This reinforces our claim that although SceneCOT **does not rely on scene tokens**, it explicitly grounds objects before answering. Since answers must be inferred solely from grounded results, the model demonstrates strong grounding–QA coherence. In contrast, Chat-Scene shows weak coherence between its QA performance and grounding performance.
>
> ### NOTE on the grounding performance of Chat-Scene:
> Although we strictly follow the authors’ instructions for building grounding data and use the official hyperparameters, we observe very low grounding performance for Chat-Scene. After extensive verification, we did not identify issues in preprocessing or training. Reports from the community and responses on the authors' GitHub repository (Issue #38) suggest that Chat-Scene is highly sensitive to training settings. Their reproduction attempts using 8 GPUs also yielded very low grounding performance. We follow the authors’ recommendation, yet performance remains low.
>
>
> To more thoroughly evaluate the grounded aspect of our chain-of-thought (CoT) reasoning, we consider both traditional grounding benchmarks and QA-grounding benchmarks.
>
> ||Nr3D(top-1 Accuracy)|Beacon3D-grounding(top1 accuracy)|MSQA-grounding(F1-score @ IoU50)|SQA3D-groundng(F1-score @ IoU 50)|ScanQA-grounding(F1-score @ IoU 50)|
> |---|---|---|---|---|---|
> |Chat-Scene|39|62.7|15.9|25.9|35.9|
> |PQ3D|**66**|**76**|10.6|10.8|19.2|
> |SceneCOT|57.7|67.8|**52.1**|**51.6**|**40.8**|
>
> In this table, we incorporate multiple grounding benchmarks to comprehensively evaluate grounding performance in both **grounding-only settings and QA-driven grounding tasks**.
> For MSQA-grounding, SQA3D-grounding, and ScanQA-grounding, the evaluation measures the model’s ability to localize the correct object instances required to answer the question. For example, given “How many chairs are on my right?”, the model must accurately ground the chairs located to the agent’s right.
>
> For SQA3D, we carefully adjust our MSQA data processing pipeline to extract object instances for all QA pairs in the test set, followed by manual verification and ID filtering to ensure data quality. For ScanQA, we directly use the official object ID annotations.
>
> The results show that **SceneCOT achieves strong and balanced grounding performance across all benchmarks**. It performs competitively on Nr3D and Beacon3D, outperforming Chat-Scene, and it significantly surpasses prior methods on MSQA-grounding, SQA3D-grounding, and ScanQA-grounding. Importantly, our model is not fine-tuned on SQA3D or ScanQA, and the strong zero-shot grounding performance further highlights the effectiveness and generalizability of our approach. The high-quality grounding outputs provide clear and reliable visual cues that support downstream reasoning.
>
> In contrast, PQ3D demonstrates strong results on Nr3D and Beacon3D but performs poorly on QA-driven grounding tasks, revealing its limitations in instruction-following scenarios. Chat-Scene shows relatively better grounding on SQA3D and ScanQA, yet still lags behind SceneCOT on the full spectrum of tasks.
>
> Overall, we believe that this comprehensive evaluation demonstrates that our framework is built upon robust and reliable grounding capabilities for both traditional grounding tasks and reasoning-oriented benchmarks.
>
> We hope these new evaluation results provide additional insights into our method. We are also working on evaluating our model on Hypo3D and VSI-Bench and will release updates once available.

---

> ### Author Response · Authors · 2025-11-28
> **Response to Reviewer rnc2 (Part 2)**
>
> ### **W2. Limited data sources**
>
> In this work, we aim to develop a transparent reasoning mechanism and introduce the first large-scale CoT dataset for 3D reasoning. SceneCOT-285K provides rich annotations for each question, including the question type, region information, and entity-level object instance annotations. In 3D scene reasoning, accurate questions and grounding texts are essential. For example, when describing or querying an object, the grounding text must uniquely refer to the intended target. Recent studies have shown that traditional benchmarks such as ScanQA contain ambiguous questions and grounding texts [1].
>
> To avoid such issues, we ensure that: the question, answer, and grounding text for each object are unambiguous.
>
> MSQA, being a situation-aware QA dataset with primary object instance IDs, naturally serves as a high-quality source for reliable reasoning. In addition, the annotation procedure of Nr3D explicitly guarantees that each grounding description uniquely corresponds to a specific object. Therefore, we select these datasets as our primary data sources.
>
> Furthermore, MMScan[2] was recently introduced as a large-scale multi-domain QA dataset for 3D scene understanding, and it also provides object ID annotations and region annotations. This dataset can benifit community for developing advanced 3D scene understanding models. We believe it can become another valuable data source for SceneCOT after applying our quality-filtering pipeline.
>
> ---
>
> ### **W3. Object-centric input**
>
> Different from previous works such as LEO, Video-3D LLM, or LLaVA-3D, SceneCOT **does not receive any scene tokens**. In other words, our model**does not take any object-centric features or voxelized scene representations as input**. As illustrated in Figure 2 (and demonstrated in the supplementary video), given a question and situational description, the model is required to first ground the object entities using the 3D grounding module based on the grounding text. It then uses the grounding results to construct task-oriented visual clues—such as object image tokens, object probabilities, or object coordinates—to support grounded reasoning.
>
> By removing all scene-token inputs, our framework **enforces a strict grounding-then-reasoning procedure**, preventing any potential shortcut from discrete scene tokens to the answer text. Instead, it focuses on the most relevant objects through explicit grounding and commonsense reasoning.

---

> ### Author Response · Authors · 2025-11-28
> **Response to Reviewer rnc2 (Part 3)**
>
> ### **Q1. Intergrating 3D-GRAND data**
>
> 3D-GRAND[3] provides a valuable data source and benchmark for studying object hallucination, and it is built upon synthetic datasets (3D-FRONT and Structured3D). The dataset also includes a scene-graph–based annotation procedure. We will further review its data instances and object annotations to assess whether they ensure strong grounding–QA coherence.
>
> In our work, we primarily focus on real-world scene understanding; therefore, we select ScanNet as our main data source. Nevertheless, we will examine the data quality of 3D-GRAND to determine whether it meets our criteria for reliable reasoning. We believe it may also serve as a useful additional resource for exploring sim-to-real transfer in 3D scene reasoning.
>
>
> ---
>
> ### **Q2. The purpose of object probability**
>
> Object probability information is essential for accurate entity grounding. For example, given the question “Which direction should I turn to get to the red-black cabinet near the door?”, if multiple cabinets exist in the scene, the model must rely on the probability scores from the 3D visual grounding module to identify which cabinet corresponds to “the red-black cabinet near the door.” These probabilities, combined with the object coordinates, enable the model to select the correct target object and determine the appropriate direction.
>
> ---
>
> ### **Q3. Effectiveness of the proposed reasoning mechanism**
>
> In Table 1, we extract the QA pairs from SceneCOT and train LEO, MSR3D, and Chat-Scene. Following their official configurations, we provide scene tokens and questions as input and supervise the models using the ground-truth answers. In contrast, SceneCOT does not use any scene tokens; instead, it learns a reasoning mechanism directly from the trajectories in the dataset.
>
> We attribute the performance drop of LEO to the differences in data mixture compared with its original training setup. The following table provides a clear illustration: LEO achieves its best performance on MSQA when trained solely on the MSQA training set. This is expected because GQA3D is predominantly an object-centric reasoning task, whereas MSQA focuses on situated reasoning. The discrepancy between these task types may lead to performance degradation when models are trained under a mixed-data setting.
>
> |Method|Training mixture|MSQA|Beacon3D|
> |---|---|---|---|
> |LEO|MSQA|54.8|--|
> |LEO|LEO-mixture|--|43.2|
> |LEO|MSQA+GQA3D|52.9|52.2|
> |SceneCOT|MSQA+GQA3D|55.6|58.9|
>
> ---
>
> We thank the reviewer again for your constructive comments and hope our response can addresse your concerns. If you still have any concern, please don't hesitate to ask!
>
> [1] 3D Question Answering for Spatial Scene Understanding. CVPR 2022
>
> [2] Scan: A Multi-Modal 3D Scene Dataset with Hierarchical Grounded Language Annotations. NeurIPS 2024
>
> [3] 3D-GRAND - A Million-Scale Dataset for 3D-LLMs. CVPR 2025

---

### Official Review · Reviewer_tikH · 2025-10-30

**Soundness:** 3
**Presentation:** 3
**Contribution:** 2
**Rating:** 6
**Confidence:** 3

**Summary:**

This paper presents SceneCoT, a 3D scene understanding framework that employs step-by-step chain-of-thought (CoT) reasoning to enhance spatial reasoning performance on benchmarks such as MSQA and Beacon3D. To facilitate 3D CoT reasoning, the authors construct a large-scale dataset, SceneCoT-185K. Experimental results further demonstrate that SceneCoT effectively improves grounding–QA coherence.

**Strengths:**

1. SceneCoT demonstrates strong performance in spatial reasoning, particularly on counting and grounding questions. Moreover, its step-wise grounded reasoning provides a transparent and interpretable rationale.

2. The construction of the CoT steps is reasonable and aligns well with how humans approach spatial question answering.

3. The paper is well-written, clearly organized, and easy to follow.

**Weaknesses:**

1. SceneCOT is designed around specific reasoning tasks—namely Situated Reasoning and Object-Centric Reasoning—and limited question types such as counting, attribute, and spatial relationship queries. This task-specific design may restrict the model’s ability to generalize to unseen question types encountered in more complex 3D world.
2. In Table 1, the overall performance of LEO and MSR3D on MSQA drops noticeably after fine-tuning on the SceneCoT-185K dataset. Could the authors provide analysis on the possible reasons behind this degradation?
3. Since SceneCOT is designed based on LLaVA-1.5, it would be helpful to show the performance comparison of these two models.
4. It would be helpful to evaluate SceneCoT on out-of-domain datasets (e.g., SQA3D [1], Hypo3D [2], VSIBench [3]) to verify whether CoT fine-tuning improves spatial reasoning beyond the training domain.

[1] Ma, Xiaojian, et al. "Sqa3d: Situated question answering in 3d scenes."

[2] Mao, Ye, et al. "Hypo3D: Exploring Hypothetical Reasoning in 3D."

[3] Yang, Jihan, et al. "Thinking in space: How multimodal large language models see, remember, and recall spaces."

**Questions:**

I have concluded all my questions in the weakness sections.

**Details Of Ethics Concerns:**

None.

---

> ### Author Response · Authors · 2025-11-28
> **Response to Reviewer tikH**
>
> ### **W1. Task generalizibility**
>
> We thank the reviewer for suggesting additional evaluations. SQA3D [1] is a widely used situated reasoning benchmark that closely matches our task domain, so we include it as an additional benchmark. Hypo3D [2] is a more recent benchmark designed to evaluate reasoning without real-time scene data. It integrates multiple existing benchmarks and emphasizes understanding scene dynamics. VSI-Bench [3] focuses on video-based spatial reasoning at the scene level.
> Since our work primarily targets situated and object-centric reasoning and proposes a method to enhance grounding–QA coherence, we add SQA3D and ScanQA as the most directly relevant additional evaluations. Nevertheless, we will discuss Hypo3D and VSI-Bench in the related work section and plan to extend our framework to both benchmarks in future work.
>
> Given time constraints, we evaluate our model on SQA3D and ScanQA in a zero-shot setting and compare it against Chat-Scene and LEO. Chat-Scene is fine-tuned on SceneCOT-grounded QA while LEO is fine-tuned on SceneCOT-QA. The results are shown below:
>
> ||SQA3D(EM-R)|SQA3D-grounding(F1-score @ IoU50)|ScanQA(EM-R)|ScanQA-grounding(F1-score @ IoU50)|
> |---|---|---|---|---|
> |Chat-Scene|36.0|3.4*|**24.5**|4.1*|
> |LEO|36.7|--|20.4|--|
> |SceneCOT|**39.7**|**51.6**|21.0|**40.8**|
>
>
> We evaluate both QA and grounding performance. The results indicate that our QA performance is comparable to Chat-Scene, while our grounding performance is substantially stronger. This reinforces our claim that although SceneCOT **does not rely on scene tokens**, it explicitly grounds objects before answering. Since answers must be inferred solely from grounded results, the model demonstrates strong grounding–QA coherence. In contrast, Chat-Scene shows weak coherence between its QA performance and grounding performance.
>
>
> ### Note on Chat-Scene’s low grounding performance:
> Although we strictly follow the authors’ instructions for building grounding data and use the official hyperparameters, we observe very low grounding performance for Chat-Scene. Reports from the community and responses on the authors' GitHub repository (Issue #38) suggest that **Chat-Scene is highly sensitive to training settings**. Their reproduction attempts using 8 GPUs also yielded very low grounding performance. Even though we follow the authors’ recommended parameters and official data preprocessing, the performance is still low.
>
> We hope these new evaluation results provide additional insights into our method. We are also working on evaluating our model on Hypo3D and VSI-Bench and will release updates once available.
>
> ---
>
> ### **W2. Performance degration of baselines**
>
> In Table 1, we extract QA pairs from SceneCOT and train LEO, MSR3D, and Chat-Scene. Following their official setups, we provide scene tokens and questions as input and supervise the models using ground-truth answers. In contrast, SceneCOT does not use scene tokens; instead, it learns reasoning patterns from trajectory data.
>
> We attribute LEO’s performance drop to differences in training data mixtures compared with its original evaluation. The table below clarifies this discrepancy. Notably, LEO achieves its best performance on MSQA when trained solely on the MSQA training set. This is expected, as **GQA3D is primarily object-centric** whereas **MSQA involves situated reasoning**, and the discrepancy between tasks likely causes degradation under mixed training.
>
> |Method|Training mixture|MSQA|Beacon3D|
> |---|---|---|---|
> |LEO|MSQA|54.8|--|
> |LEO|LEO-mixture|--|43.2|
> |LEO|MSQA+GQA3D|52.9|52.2|
> |SceneCOT|MSQA+GQA3D|55.6|58.9|
>
> ---
>
> ### **W3. Performance of LLaVA-1.5**
>
> We wish to clarify that our work focuses on **3D scene reasoning**, which requires comprehensive understanding of an entire 3D environment. LLaVA-1.5, however, is primarily trained on **single-image** VQA datasets and therefore cannot capture whole-scene 3D structure effectively. In contrast, our framework processes a full 3D point cloud and multiple images. LLaVA-1.5 is used only as a sub-module to handle attribute and description queries during the grounded reasoning stage.
>
> ---
>
> ### **W4. Evaluation**
>
> Please refer to our response to Weakness 1 (Section W1).
>
> ---
>
> We thank again for the reviewer's careful review and valuable suggestions for our work and hope our response can solve your concerns. If you have any further concern, please let us know!
>
>
> [1] SQA3D: Situated Question Answering in 3D Scenes. ICLR 2023
>
> [2] Hypo3D: Hypo3D: Exploring Hypothetical Reasoning in 3D. ICML 2025
>
> [3] VSI-Bench: Thinking in Space: How Multimodal Large Language Models See, Remember, and Recall Spaces. CVPR 2025

---

### Official Review · Reviewer_WaKT · 2025-10-30

**Soundness:** 3
**Presentation:** 3
**Contribution:** 3
**Rating:** 4
**Confidence:** 4

**Summary:**

This paper addresses the problem of poor grounding in 3D vision-language models, where models often generate plausible-sounding answers that are not factually connected to the 3D scene. The authors propose SCENECOT, a novel framework that introduces step-by-step, Chain-of-Thought (CoT) reasoning to 3D question answering. The method explicitly decomposes a complex 3D reasoning task into four manageable stages: task recognition, task-relevant region localization, entity/attribute grounding using expert modules, and final grounded reasoning. To train this framework, the authors also developed SCENECOT-185K, the first large-scale dataset containing 185,000 grounded CoT reasoning traces for 3D scenes. Experiments demonstrate that SCENECOT achieves competitive performance on the general MSQA benchmark and, most notably, significantly outperforms all baselines on the Beacon3D benchmark, which is specifically designed to measure grounding-QA coherence.

**Strengths:**

1. This work directly targets the critical and well-documented problem of poor grounding-QA coherence in 3D-VL models. Instead of just aiming for better QA accuracy, it focuses on ensuring the answers are correctly derived from the scene's visual context.
2. The framework's multi-stage design (task recognition, region localization, grounding, reasoning) is intuitive and inherently interpretable. This transparency makes it easier to diagnose failure cases, as shown in the qualitative examples.
3. The paper is well-written and easy to follow.

**Weaknesses:**

1. The SCENECOT framework is a complex, multi-stage pipeline rather than a simple end-to-end model. Its performance is heavily reliant on a cascade of specialized, pre-trained modules (e.g., Mask3D for object proposals, PQ3D for grounding). This introduces multiple potential points of failure, and the overall performance is strongly coupled to the quality of these "expert" modules.
2. The dataset, while large, is constructed from existing benchmarks (MSQA, Nr3D) that are primarily based on the ScanNet dataset. This limits the diversity of scenes, objects, and tasks. The paper acknowledges that the framework does not yet extend to more complex, long-horizon embodied tasks.
3. The paper introduces a "grounded CoT" framework, but it lacks experiments on standard 3D visual grounding benchmarks (e.g., Nr3D, Sr3D, or ScanRefer). While Beacon3D measures coherence (grounding + QA), evaluating the grounding module's performance in isolation on these tasks seems essential to fully validate the "grounded" aspect of the CoT.

**Questions:**

1. The generation of the SCENECOT-185K dataset relies on rule-based methods and LLM (GPT-4O) generation. What was the extent of manual verification to ensure the quality and correctness of the intermediate reasoning "thoughts"?
2. In the 4-stage pipeline, how does the model handle error propagation? For instance, if the initial "Task Recognition" step fails, or the "Region Localization" identifies the wrong area, does this inevitably lead to a final failure, or are there mechanisms for recovery in the later stages?
3. The multi-step inference process, which involves calls to symbolic engines and expert grounding modules, seems computationally more intensive than an end-to-end model. What is the comparative inference latency of SCENECOT versus baselines like Chat-Scene or LLaVA-3D? Is the framework practical for real-time applications?

---

> ### Author Response · Authors · 2025-11-28
> **Response to Reviewer WaKT (Part 1)**
>
> Thanks for your valuable feedback and constructive comments. We appreciate the positive recognition of our work's strengths, including good motivation, novel method, and dataset contribution. We will address the reviewer's concerns as below.
>
> ---
>
> ### **W1. Multi-stage pipeline**
>
> Our framework is not a simple cascade-based, inference-only pipeline. The 3D visual grounding module is jointly optimized with the full loss function (see Line 212), enabling it to learn to ground target objects from the grounding text enclosed by “<grd> … </grd>”. The other components—such as Mask3D, region selection, and the object-relative coordinate computation functions—remain fixed during training (see definitions in Appendix A).
>
> We emphasize that our framework **not only inherits the capabilities of the expert modules but also benefits from a human-like, step-by-step reasoning chain**. For instance, SceneCOT can effectively handle challenging scenarios, including SQA3D and SQA3D-grounding (zero-shot). Please refer to the grounding evaluation results in Weakness 3 for details. The strong grounding performance is largely attributed to our well-designed reasoning chain. We first generate “region-text” and then programmatically localize objects within each sub-region. This filtering mechanism simplifies the grounding process and enables more accurate identification of target objects in situated reasoning settings.
>
> Furthermore, apart from the grounding module and semantic extractor, our framework maintains high accuracy in stage-1 and stage-2 predictions. Both task recognition and region recognition are highly reliable, providing strong guidance for subsequent reasoning steps. These results are also discussed in our explanation for Weakness 3.
>
> ---
>
> ### **W2. Dataset domains**
>
> In this work, we focus on developing a reliable visually grounded reasoning mechanism for 3D scene understanding and achieve strong grounding–QA coherence. Our primary emphasis is on object-centric and situated reasoning tasks. To further validate the effectiveness of our method, we additionally provide evaluation results on ScanQA and SQA3D in zero-shot settings. Extending our step-by-step reasoning framework to more complex, long-horizon embodied tasks represents a promising direction for future research.
>
> ---
>
> ### **W3. Evaluation of grounding tasks**
>
> We appreciate the reviewer’s valuable suggestion. To more thoroughly evaluate the grounded aspect of our chain-of-thought (CoT) reasoning, we consider both traditional grounding benchmarks and QA-grounding benchmarks.
>
> ||Nr3D(top-1 Accuracy)|Beacon3D-grounding(top1 accuracy)|MSQA-grounding(F1-score @ IoU50)|SQA3D-groundng(F1-score @ IoU 50)|ScanQA-grounding(F1-score @ IoU50)|
> |---|---|---|---|---|---|
> |Chat-Scene|39|62.7|15.9|25.9|35.9|
> |PQ3D|**66**|**76**|10.6|10.8|19.2|
> |SceneCOT|57.7|67.8|**52.1**|**51.6**|**40.8**|
>
> In this table, we incorporate multiple grounding benchmarks to comprehensively evaluate grounding performance in both **grounding-only settings and QA-driven grounding tasks**.
> For MSQA-grounding, SQA3D-grounding, and ScanQA-grounding, the evaluation measures the model’s ability to localize the correct object instances required to answer the question. For example, given “How many chairs are on my right?”, the model must accurately ground the chairs located to the agent’s right.
>
> For SQA3D, we carefully adjust our MSQA data processing pipeline to extract object instances for all QA pairs in the test set, followed by manual verification and ID filtering to ensure data quality. For ScanQA, we directly use the official object ID annotations.
>
> The results show that **SceneCOT achieves strong and balanced grounding performance across all benchmarks**. It performs competitively on Nr3D and Beacon3D, outperforming Chat-Scene, and it significantly surpasses prior methods on MSQA-grounding, SQA3D-grounding, and ScanQA-grounding. Importantly, our model is not fine-tuned on SQA3D or ScanQA, and the strong zero-shot grounding performance further highlights the effectiveness and generalizability of our approach. The high-quality grounding outputs provide clear and reliable visual cues that support downstream reasoning.
>
> In contrast, PQ3D demonstrates strong results on Nr3D and Beacon3D but performs poorly on QA-driven grounding tasks, revealing its limitations in instruction-following scenarios. Chat-Scene shows relatively better grounding on SQA3D and ScanQA, yet still lags behind SceneCOT on the full spectrum of tasks.
>
> Overall, we believe that this comprehensive evaluation demonstrates that our framework is built upon robust and reliable grounding capabilities for both traditional grounding tasks and reasoning-oriented benchmarks.

---

> ### Author Response · Authors · 2025-11-28
> **Response to Reviewer WaKT (Part 2)**
>
> ### **Q1. Data quality**
>
> We manually reviewed the accuracy of the target object IDs reported in Table 7 of Appendix B. For clarity, we present the results again below:
>
> #### **Accuracy of target object IDs (Table 7 in Appendix B)**
>
> ||counting|spatial relationship|refer|attribute|navigation|overall|
> |---|---|---|---|---|---|---|
> |#sampled instances|50|50|50|50|50|250|
> |accurracy|92%|100%|98%|100%|100%|98% |
>
> |Sampled instances|Quality|
> |---|---|
> |100|100|
>
> The object instance IDs used in our dataset all lie within the object regions we constructed. The results in Table 7 confirm that the object instance information is highly accurate and that our region selection rules are reliable and well-defined.
>
> For question-type annotations, we follow the official question types provided in MSQA. The grounding text for MSQA is generated using GPT-4o. To ensure quality, we manually verified whether each piece of grounding text accurately reflects the relevant object descriptions for its corresponding QA pair. For example, given the question “How many red chairs are in the room?”, the expected grounding text should be “the red chairs in the room”. The results shown in the table above demonstrate that the grounding text quality is consistently high.
>
> For GQA3D, all QA pairs are generated directly from the target object images, resulting in 100% accuracy for target object IDs.
>
> ---
>
> ### **Q2. Sub-stage accuracy**
>
> In our framework, the errors from the two stages—Task Recognition and Region Localization—can be largely ignored due to their high accuracy. Below, we provide typical examples for the question:
>
> Question: “How many chairs are on my right?”
>
> ||Question type recognition|Region recognition|
> |---|---|---|
> |Correct|Counting|Now I need to ground the objects on my right.|
> |Wrong|Attriute|Now I need to ground the objects on my left.|
>
> ||Question type recognition|Region recognition|
> |---|---|---|
> |Accuracy|99.4|100.0|
>
> These results show that both sub-tasks achieve extremely high accuracy, and thus the resulting errors are negligible in practice. We attribute this reliability to the strong commonsense priors inherent in large language models.
>
> For Region Localization, we use a rule-based filtering mechanism described in Appendix A.3. Given a region instruction such as “Now I need to ground the objects on my right.”, the system strictly applies the predefined rules to filter objects located within the corresponding sub-region. This ensures consistent and accurate region-based object selection aligned with the model’s region recognition results.
>
> ---
>
> ### **Q3. Inference latency**
>
> Our framework primarily focuses on enabling a **reliable and transparent reasoning process**, which requires incorporating all necessary contextual information during inference. As a result, it inevitably incurs higher inference time due to module calls and the construction of detailed reasoning context. Below, we provide a qualitative comparison:
>
> |Method|Object instance information?|Reasoning related token length|Inference latency|
> |--|--|--|--|
> |LEO|No|0|4.8s|
> |LLaVA-3D|No|0|0.2s|
> |Chat-Scene|No|0|0.5s|
> |SceneCOT|Yes|350~1500|10.4s|
>
> |Overall|Stage 1-3 sequence generation|Stage 4 sequence generation|Visual clue construction|
> |--|--|--|--|
> |10.4s|4.2s|3.8s|2.4s|
>
> We compare the reasoning-token length and inference latency of SceneCOT with LEO, LLaVA-3D, and Chat-Scene. Since these baseline models only generate a final answer without producing intermediate rationales, their latency primarily consists of a single forward pass of sequence generation.
>
> In contrast, our framework provides **rich intermediate reasoning context**, including question type, object instances, and their spatial grounding in the scene. This requires both additional information gathering and multiple rounds of sequence generation based on the constructed visual clues.
>
> Our analysis shows that the majority of the latency comes from LLM sequence generation, while the time spent on visual clue construction is relatively moderate. Overall, we believe this represents a **reasonable trade-off between transparent, interpretable reasoning and inference-time latency**.
>
> ---
>
> We sincerely thank the reviewer again for the careful assessment and constructive suggestions. We hope our explanations and additional experiments adequately address your concerns, and we welcome any further questions.

---

### Official Review · Reviewer_rALF · 2025-11-01

**Soundness:** 4
**Presentation:** 3
**Contribution:** 3
**Rating:** 6
**Confidence:** 3

**Summary:**

1. The paper proposes SCENECOT, a grounded Chain-of-Thought (CoT) framework for interpretable 3D scene reasoning. It decomposes complex reasoning into four explicit steps—task recognition, region localization, entity grounding, and grounded reasoning—each supported by symbolic and multimodal expert modules.

2. It introduces SCENECOT-185K, a dataset of 185K stepwise reasoning traces covering Situated (MSQA) and Object-Centric (Beacon3D, GQA3D) tasks.

3. Experiments on MSQA and Beacon3D demonstrate improved grounding–QA coherence (34.7% vs. 19.5%) and validated gains from question-type recognition, region filtering, and grounding loss.

**Strengths:**

1. Novel application of CoT in 3D reasoning: Introduces an interpretable step-by-step framework that explicitly grounds each reasoning stage in scene elements. Visualization of reasoning chains (Fig. 6) makes the model’s decision process transparent and easier to diagnose.

2. Large-scale dataset: SCENECOT-185K is the first dataset pairing CoT traces with 3D scene data, supporting both situated and object-centric reasoning.

3. Clear empirical validation: Comprehensive experiments (MSQA + Beacon3D) and ablations demonstrate consistent improvement in 3D VQA task.

**Weaknesses:**

1. Lack of Human Verification. All QA pairs in the SCENECOT-185K dataset are generated by GPT-4o, which can introduce factual or logical errors. The paper does not mention any human validation or quality control process to ensure the correctness of the generated reasoning traces.

2. Limited Baseline Coverage. Although GPT-4o is included as a comparison model, the experiments omit stronger recent multimodal baselines such as Gemini 2.5 Pro or Claude Opus, which would provide a more comprehensive evaluation of reasoning capability.

**Questions:**

1. In the Grounded Reasoning, how is the "Object Image Tokens" generated? Are they derived from cropped RGB patches, projected 3D features, or another visual encoding process?

2. What exactly is the symbolic engine mentioned in the architecture?

---

> ### Author Response · Authors · 2025-11-28
> **Response to Reviewer rALF**
>
> We greatly appreciate the reviewer’s careful assessment and insightful comments. We respond to the concerns as follows.
>
> ---
>
> ### **W1. Human Verification**
>
> We have mannually reviewed the accuracy of the target object IDs in the Table 7 in Appendix B. Here we represent the results as follows again:
>
> Accuracy of target object IDs (Table 7 in Appendix B)
>
> ||counting|spatial relationship|refer|attribute|navigation|overall|
> |---|---|---|---|---|---|---|
> |#sampled instances|50|50|50|50|50|250|
> |accurracy|92%|100%|98%|100%|100%|98% |
>
> |Sampled instances|Quality|
> |---|---|
> |100|100|
>
> The object instance IDs are within the object region we built. Table 7 shows that the quality of object instance information is accurate and region selection rule is also concrete. While for the question type information, we follow the offical quesiton type in MSQA. The grounding text for MSQA in our dataset is generated by GPT-4o. To guarantee the quality, we also checked the quality of the "grounding text" to check if it can reflect the object textual information for the QA pair. For example, the quesiton "How many red chairs are in the room?". The grounding text should be "the red chairs in the room". The results in above table shows that the grounding text quality is high.
>
> ---
>
> ### **W2. Limited Baseline Coverage**
>
> In this paper, we mainly focus on fine-tuned models and we compare MLLMs like GPT-4o as a reference since they recieve more input such as ground-truth object locations in MSQA evaluation. It is regretful that we could not access to Gemini 2.5-Pro and Claude Opus owing to region restriction. However, we provide more evaluation results of grounding and QA benchmarks.
>
> To provide additional evaluation, we assess our model on SQA3D and ScanQA under zero-shot settings and compare it with Chat-Scene and LEO. Both Chat-Scene and LEO are fine-tuned on SceneCOT-QA: LEO is fine-tuned on the SceneCOT-QA dataset, while Chat-Scene is fine-tuned on the SceneCOT-grounded QA variant, where the model must first predict the target object IDs before generating the answer. The results are as follows:
>
> Given time constraints, we evaluate our model on SQA3D and ScanQA in a zero-shot setting and compare it against Chat-Scene and LEO. Chat-Scene is fine-tuned on SceneCOT-grounded QA while LEO is fine-tuned on SceneCOT-QA. The results are shown below:
>
> ||SQA3D(EM-R)|SQA3D-grounding(F1-score @ IoU50)|ScanQA(EM-R)|ScanQA-grounding(F1-score @ IoU50)|
> |---|---|---|---|---|
> |Chat-Scene|36.0|3.4*|**24.5**|4.1*|
> |LEO|36.7|--|20.4|--|
> |SceneCOT|**39.7**|**51.6**|21.0|**40.8**|
>
>
> We evaluate both QA and grounding performance. The results indicate that our QA performance is comparable to Chat-Scene, while our grounding performance is substantially stronger. This reinforces our claim that although SceneCOT **does not rely on scene tokens**, it explicitly grounds objects before answering. Since answers must be inferred solely from grounded results, the model demonstrates strong grounding–QA coherence. In contrast, Chat-Scene shows weak coherence between its QA performance and grounding performance.
>
> ### Note on Chat-Scene’s low grounding performance:
> Although we strictly follow the authors’ instructions for building grounding data and use the official hyperparameters, we observe very low grounding performance for Chat-Scene. Reports from the community and responses on the authors' GitHub repository (Issue #38) suggest that **Chat-Scene is highly sensitive to training settings**. Their reproduction attempts using 8 GPUs also yielded very low grounding performance. Even though we follow the authors’ recommended parameters and official data preprocessing, the performance is still low.
>
> ---
>
> ### **Q1. The details of object image tokens**
>
> After 3D visual grounding procedure, we get the object instance, then we can project the 3D instance to RGBD frames and fetch the correponding image that contain the target 3D instance. Then we feed the object images to our MLLM visual encoder and extract the image tokens.
>
> ---
>
> ### **Q2. The details of the symbolic engine**
>
> We provide the definations of symbolic engines in Appendix A.2, including **Spatial Region Recognition** and **Visual Clue Construction**. Spatial region recognition module partitions the environment into sub-regions based on the agent’s location and orientation. Visual clue construction module generates textual visual clues by integrating object probabilities, semantic labels, and spatial locations.
>
> We thank the reviewer again for your constructive comments and hope our response can addresse your concerns. If you still have any concern, please don't hesitate to ask!

---

### Meta-Review · Area_Chair_iqTY · 2026-01-06

**Summary:**

This paper proposes **SCENECOT**, a framework for eliciting grounded Chain-of-Thought (CoT) reasoning in 3D scenes. The core idea is to decompose complex 3D reasoning tasks into four explicit steps: task recognition, region localization, entity grounding, and grounded reasoning. This process is supported by a modular architecture involving an MLLM and specialized expert modules (e.g., Mask3D). To facilitate this, the authors construct **SCENECOT-185K**, a large-scale dataset of ~185k reasoning traces derived from MSQA and Nr3D. The key contribution lies in enforcing explicit object grounding within the reasoning chain to improve interpretability and grounding-QA coherence. Experiments on MSQA and Beacon3D benchmarks show that SCENECOT achieves competitive QA accuracy while significantly outperforming baselines in grounding consistency.

**Reviewer Concerns:**

Reviewers raised several valid concerns, primarily regarding **data quality**, **evaluation breadth**, **pipeline complexity**, and **generalization**.
- **rALF** and **WaKT** questioned the lack of human verification for GPT-4o-generated reasoning traces. The authors responded with manual checks showing high accuracy (≈98–100%) for object IDs and grounding texts.
- **WaKT**, **tikH**, and **rnc2** noted limited evaluation on only MSQA and Beacon3D. In rebuttal, the authors provided new zero-shot results on SQA3D and ScanQA, showing competitive QA and superior grounding performance.
- **WaKT** and **rnc2** expressed concern about the multi-stage pipeline’s dependence on fixed modules (e.g., Mask3D) and its inference latency. The authors clarified that only region selection and Mask3D are fixed; the grounding module is jointly trained. They acknowledged higher latency but argued it is a worthwhile trade-off for interpretability.
- **tikH** and **rnc2** questioned generalizability to other benchmarks (e.g., Hypo3D, VSI-Bench) and data sources. The authors explained that their focus is on situated and object-centric reasoning with reliable annotations, but they plan to extend to broader benchmarks in future work.

Overall, the authors addressed most concerns with additional experiments, clarifications, and manual verification, strengthening the paper's credibility.

**Reviewer Scores:**

The reviews reflect a mix of positive reception for the idea and cautiousness about the system's complexity and evaluation breadth.
*   **Reviewer rALF:** Score 6 (Marginally above acceptance). *Praised the novelty and dataset; concerned about lack of human verification (addressed in rebuttal).*
*   **Reviewer WaKT:** Score 4 (Marginally below acceptance). *Appreciated the focus on grounding but concerned about the complex pipeline and latency.*
*   **Reviewer tikH:** Score 6 (Marginally above acceptance). *Found the CoT construction reasonable and effective; requested out-of-domain evaluation (addressed in rebuttal).*
*   **Reviewer rnc2:** Score 4 (Marginally below acceptance). *Acknowledged leading performance but critiqued the limited evaluation benchmarks and object-centric input reliance.*

Two reviewers marginally favor acceptance (6), while two marginally favor rejection (4). Confidence levels are moderate to high, with rnc2 being “absolutely certain.” The authors’ detailed rebuttal appears to have addressed several key concerns, particularly around evaluation and data quality.

---

### Decision · Program_Chairs · 2026-01-26

Accept (Poster)